# Tastes and retronasal odours evoke a shared flavour-specific neural code in the human insula

Putu Agus Khorisantono [1] ✉, Maria G. Veldhuizen [2,3,4,5] & Janina Seubert [1] ✉

During food consumption, tastes combine with retronasal odours to form flavour, which leads to a link so robust that retronasal odours can elicit taste sensations without concurrent taste stimulation. However, the cortical integration of these parallel sensory signals remains unclear. Here, we combine a flavour-binding paradigm and functional neuroimaging to test whether retronasal odorants evoke encoding patterns in the insula similar to those of their paired tastants. Healthy participants attend a familiarisation session with congruent sweet and savoury flavours followed by two functional MRI (Magnetic Resonance Imaging) sessions where they separately receive the constituent tastants and odorants. Multivariate pattern analysis reveal classification of retronasal odours within the insula, exhibiting overlapping representations with their associated tastes, particularly in the ventral anterior insula. Additionally, we observe temporal drift in insular taste representations, paralleling findings in rodent gustatory cortex. These findings underscore the robust crossmodal influences of gustatory and retronasal olfactory processing that underpin the flavour percept.

Most of us who have suffered from a cold can attest to the illusory feeling of taste loss that comes with a blocked nose. This happens because normally, when food is in the mouth without a blocked nose, odorants reach the olfactory epithelium through the back of the throat (retronasal olfaction) as tastants elicit a concurrent taste sensation (Fig. 1A). Both sensations are then perceived as a holistic flavour identity. Indeed, so strong is the quasi-synaesthetic relationship between taste and odour that retronasal odours not only enhance taste perception[1,2] but can even elicit a taste percept in the absence of a tastant[3]. For example, strawberry aroma in the absence of sweet receptor stimulation still 'tastes' distinctly 'sweet'. Moreover, words used to describe odours tend to either refer to their source or an associated taste, which implies that odours are conceptually mapped to objects or tastes[4,5]. While verbal representations of odours might overlap with tastes, it remains poorly understood to what extent these canonically parallel pathways overlap in the central nervous system.

The primary sensory cortex involved in the processing of odours has been localised to the piriform cortex in a range of mammalian species. There is a level of preferential processing in the piriform, where the anterior piriform cortex encodes the structure of an odorant and the posterior piriform cortex encodes the identity[6]. Neural encoding of tastants, on the other hand, has highlighted the insula as the primary gustatory cortex, shown in rodents[7], non-human primates[8] and humans[9,10]. Curiously, neurons in the gustatory cortex are also known to respond to odours[11–13]. Recent literature has shown reliably distinct ensemble encoding of retronasal odours in the rodent gustatory cortex[14]. Olfactory-induced activations in the insula are unexpected as, canonically, information from primary sensory cortices is processed in parallel before integration by the orbitofrontal cortex

[1]Division of Psychology, Department of Clinical Neuroscience, Karolinska Institutet, SE-171 77, Stockholm, Sweden. [2]Department of Psychology, Faculty of Humanities and Social Sciences, Mersin University, Mersin, Türkiye. [3]Department of Anatomy, Faculty of Medicine, Mersin University, Mersin, Türkiye. [4]Aysel Sabuncu Brain Research Center (ASBAM), Bilkent University, Ankara, Türkiye. [5]National Magnetic Resonance Research Center (UMRAM), Bilkent University, Ankara, Türkiye. ✉e-mail: putu.agus.khorisantono@ki.se; janina.seubert@ki.se

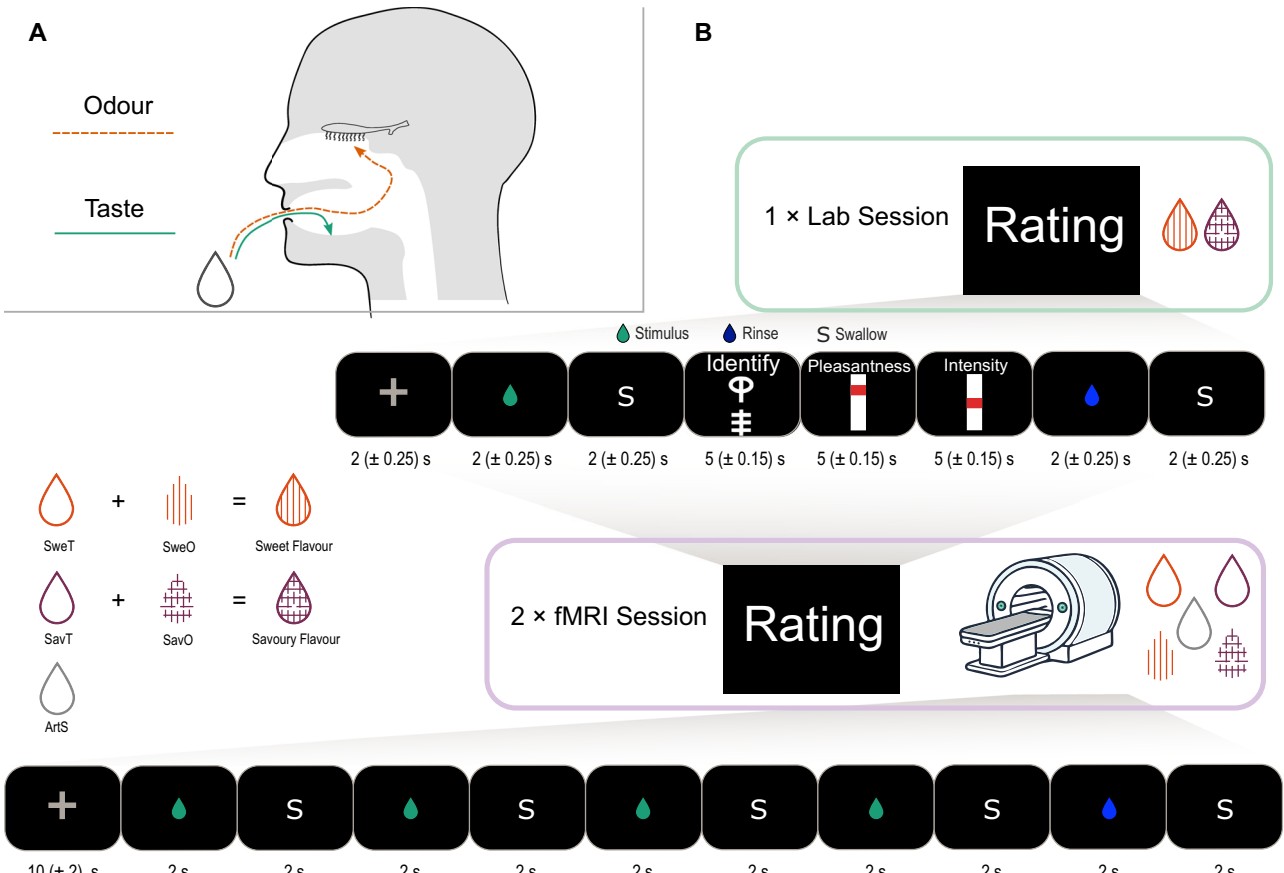

**Fig. 1 | Pathways of chemosensation in the oral cavity and experimental paradigm. A** Different routes for taste and odour during food consumption. While the flavour percept is holistic, it relies on chemosensation through parallel trans-duction routes. **B** Setup of the experiment. Participants attend one laboratory session where they are familiarised with the sweet and savoury flavour mixtures. In two subsequent sessions, they undergo fMRI while receiving unimodal stimuli. SweT: sweet taste; SweO: sweet odour; SavT: savoury taste; SavO: savoury odour; ArtS: artificial saliva.

(OFC), which encodes the identity[8,15] and subjective value[15–17] of a food item. However, gustatory and olfactory stimulation occur almost simultaneously and with great contingency during food consumption, likely encouraging Hebbian pairing between neurons responsive to both chemosensory signals. Both behavioural[3,18,19] and functional neuroimaging data[20,21] show the existence of taste-odour associations that imply a strong link between these chemosensory signals. Over-lapping representations of smells and tastes may occur prior to the interaction of the chemosensory percept with other sensory proper-ties of foods in the OFC. In non-human primates, monosynaptic pro-jections from the piriform olfactory cortex to the dysgranular and agranular insula are well-documented[22,23] and constitute a putative pathway for olfactory-gustatory integration during food flavour per-ception. We therefore hypothesise that a similar pathway is leveraged in humans, where olfactory information is integrated with gustatory information in the insula to form a holistic flavour concept to induce taste-like activations to odours with prior taste associations. Addi-tionally, ensemble taste representations in the rodent gustatory cortex are known to shift with repeated exposure[24,25]. This key feature, along with encoding of retronasal odours in the insula, may also be pre-served in humans.

Here, we use oral delivery of unimodal chemosensory stimulation (tasteless odorants and odourless tastants; Fig. 1A) and functional magnetic resonance imaging (fMRI) to investigate early overlap in processing of chemosensory information in the insula. Specifically, we test if retronasal taste-associated odours elicit dissociable patterns of activation in the gustatory cortex and if these patterns overlap with those of the associated tastes. We show that familiar taste-odour pairs

share overlapping patterns in a taste-responsive region of the insula. Crucially, after parcellating the insula based on layer morphology (granularity), we show that this crossmodal overlap is observed mostly in the ventral anterior subregion of the insula, which corresponds to areas of lower granularity, aligning with the abovementioned mono-synaptic piriform-insular projections[22,23]. Further analyses to char-acterise temporal fluctuations in taste and flavour representations showed instability of taste identity patterns, which, to our knowledge, had not been previously shown in humans.

## Results
### Task design
We tested the pre-registered hypothesis of taste-like activation pat-terns in response to retronasal odour using fMRI in 25 participants (11 male, 14 female by self-report) while they orally received unimodal taste or odour stimuli. Participants attended a laboratory session where they received congruent flavours (a sweet solution with a sweet-associated aroma [henceforth referred to as sweet odour] and a savoury solution with a savoury-associated aroma [henceforth referred to as savoury odour]) while performing identification and rating tasks. In two subsequent separate scanning sessions, they underwent fMRI during a passive tasting task where they orally received the unimodal components of the previously exposed flavour stimuli (odourless tastants or tasteless odorants—see Fig. 1B). Prior to each scanning session, participants also rated the pleasantness and intensity of each stimulus and performed the identification task again to ensure that they were indeed able to differentiate the sweet and savoury stimuli. Supplementary Fig. 2A shows the distribution of the odours used. As

seen in Supplementary Fig. 2B, C, we observed intra- and interpersonal variability in perceived intensity and pleasantness ratings within each of the stimulus categories. However, these differences did not vary systematically between stimulus categories and thus did not cause any systematic differences between them at the group level. Supplementary Fig. 3 shows the session-wise identification accuracy per stimulus. A linear mixed-effects model (LMM) of the accuracy showed neither main effects of session or flavour nor interaction effects of the same (Supplementary Table 1).

### Specific sensory cortex activation in response to modality-specific stimulation

We first used mass-univariate General Linear Model (GLM) analyses of modality-specific stimulation, contrasted against the artificial saliva (ArtS) control, to test which regions show activation in response to chemosensory stimulation. As seen in Fig. 2A and Supplementary Table 2, oral delivery of tasteless odorants induced whole-brain corrected significant BOLD activations in the left piriform cortex ([−18, −2, −14], peak z = 5.56, $p_{FWE}$ = .022), which is part of the primary olfactory cortex. As the piriform cortex was a pre-registered a priori region of interest, we used a small-volume correction (SVC) based on statistical localisation by activation likelihood estimation[26] to show significant activation in the right piriform cortex ([26, 0, −16], peak z = 4.17, $p_{FWE-SVC}$ = .005) in response to retronasal odour stimulation.

Activations elicited by odourless tastants contrasted against the ArtS control peaked in the bilateral mid-dorsal insula ([−36, −10, 10], peak z = 5.16, $p_{FWE}$ = .004; [36, −6, 12], peak z = 4.66, $p_{FWE}$ = .018), consistent with activation of canonical primary taste cortex. Notably, the cluster in the left hemisphere extended to parts of the ventral anterior insula (Fig. 2A and Supplementary Table 2), and a significant cluster was seen in the right ventral anterior insula after SVC ([−8, 4, 10], peak z = 4.46, $p_{FWE-SVC}$ = .010).

### Identity-specific decoding of chemosensory stimulus quality in the insula

In order to test for the presence of identity-specific representations of chemosensory stimuli in the insula beyond modality-specific univariate activations, we performed a pre-registered multivariate pattern analysis (MVPA) decoding in independent functional ROIs in the bilateral insula responsive to tastants, generated using leave-one-subject-out cross-validation (see "Methods" and Supplementary Fig. 4). We generated a GLM with quality-specific regressors for odours and tastants, i.e., one regressor each for sweet taste (SweT), savoury taste (SavT), sweet odour (SweO), savoury odour (SavO) and artificial saliva (ArtS) and trained a support vector machine (SVM) on the resultant betas from all runs but one (leave-one-run-out cross-validation, see "Methods"). As hypothesised, the classifier could differentiate sweet and savoury taste (mean accuracy of 57.31%, permutation test $p = 9 \times 10^{-4}$), confirming that taste-responsive regions in the bilateral insula show dissociable patterns of activation in response to different tastants (Fig. 2B). Notably, odour quality could similarly be decoded (mean accuracy of 56.67%, permutation test $p$ = .003), indicating that this taste-responsive region also shows differential activation to taste-associated odours. We subsequently tested whether patterns of taste-associated odours in the primary gustatory cortex overlap with those of their associated tastes by training a classifier on taste quality and testing it on odour quality and vice versa. Indeed, this crossmodal classifier decoded taste or odour quality significantly above chance (mean accuracy = 53.83%, permutation test $p$ = .009), indicating that flavour-specific insular representations of flavour quality are conserved across stimulus modalities. To rule out the possibility that these decoding accuracies were driven by hedonic differences, we compared the resultant crossmodal accuracy against absolute differences in hedonic ratings of each participant between sweet and savoury

stimuli. We found no correlation between the hedonic differences and crossmodal decoding performance (Fig. 2E).

The functional ROI generated from the leave-one-subject-out method included both dorsal mid-insula and ventral anterior insula, which extend beyond areas associated with primary gustatory cortex. These range from granular to agranular regions and are slightly different for each subject (Supplementary Fig. 4). Given that insular projections from the piriform cortex largely terminate in its dysgranular and agranular sections in both primates and rodents[23,27], we suspected that our functional ROI captured functionally separate parcellations of the insular cortex. In an exploratory follow-up analysis, we tested for differential identity-specific responsiveness to flavour components in sub-regions parcellating the ROI based on granularity (Fig. 2C) using a probabilistic anatomical map[28] to show the specific sub-regions showing overlapping patterns for tastes and odours. The granular insula (Fig. 2D) displayed reliable taste decoding with a mean accuracy of 59.22% ($p = 2 \times 10^{-5}$), but no significant decoding of odours (mean accuracy 52.67%, $p$ = .127) or crossmodal decoding of flavour identity (mean accuracy = 50.84%, $p$ = .308). On the other hand, dysgranular and agranular anterior insula (Fig. 2D) activation patterns reflected tastant identity (mean accuracy 60.06%, $p = 8 \times 10^{-6}$), odour identity (mean accuracy 56.35%, $p$ = .004) and crossmodal flavour-specific patterns between the two modalities (mean accuracy 57.78%, $p < 1 \times 10^{-6}$, uncorrected one-tailed comparison to permuted null distribution for all). Critically, the dysgranular and agranular ROI displayed better accuracy for the crossmodal decoder than the granular insula (t(24) = 3.028, $p$ = .006, $d_{rm}$ = 0.626, 95% CI = [0.022, 0.116]; Fig. 2D), but there was no significant difference between the ROIs for the taste (t(24) = 0.292, $p$ = .760, $d_{rm}$ = 0.061, 95% CI = [-0.051, 0.068]) and odour decodability (t(24) = 1.288, $p$ = .195, $d_{rm}$ = 0.267, 95% CI = [-0.023, 0.096], uncorrected two-tailed permutation-based t-tests). Supplementary analyses of taste, odour and crossmodal decoding in each of the 12 probabilistic ROIs in the same atlas can be seen in Supplementary Fig. 5.

In order to ensure that the above-chance decoding we observed in our ROIs are indeed reflective of overlapping activation patterns, we extracted the tuning index for each voxel to each modality by contrasting their statistical parametric maps (see "Methods" and Fig. 3A). We found that tuning index overlap, derived from crossmodal correlations of tuning index maps, were predictive of the performance of the crossmodal decoder in the leave-one-subject-out gustatory ROI (r = .507, $p$ = .010; Fig. 3B), as well as the anatomically defined parcellations of the granular insula (r = .557, $p$ = .004; Fig. 3C) and dysgranular and agranular insula (r = .535, $p$ = .006; Fig. 3D). This indicates that above-chance crossmodal decoding was indeed driven by overlapping flavour-specific patterns of activations between the modalities. These results highlight the insula's unique role in flavour perception, starting from the processing of basic tastant quality in the granular insula and its integration into a shared representation with odour quality that likely underlies the formation of a flavour percept in the dysgranular and agranular insula.

### Whole-brain population-based decoding of chemosensory stimulus quality

In order to investigate whether any region outside the insula displays shared patterns between flavour components, we performed a whole-brain searchlight analysis with our crossmodal MVPA. Figure 4 shows clusters that survive threshold-free cluster enhancement (TFCE) of $10^4$ permutations. We were able to decode taste from odour and odour from taste in a region of the ventral anterior insula that slightly overlaps but is anterior to the taste-responsive ROI (local peak z = 1.87, $p_{TFCE}$ = .031), the middle frontal gyrus extending into the lateral orbitofrontal cortex (lOFC; local peak z = 2.11, $p_{TFCE}$ = .018) and the medial orbitofrontal cortex (mOFC; local peak z = 1.68, $p_{TFCE}$ = .047). However, the highest crossmodal decoder performance was observed in a region

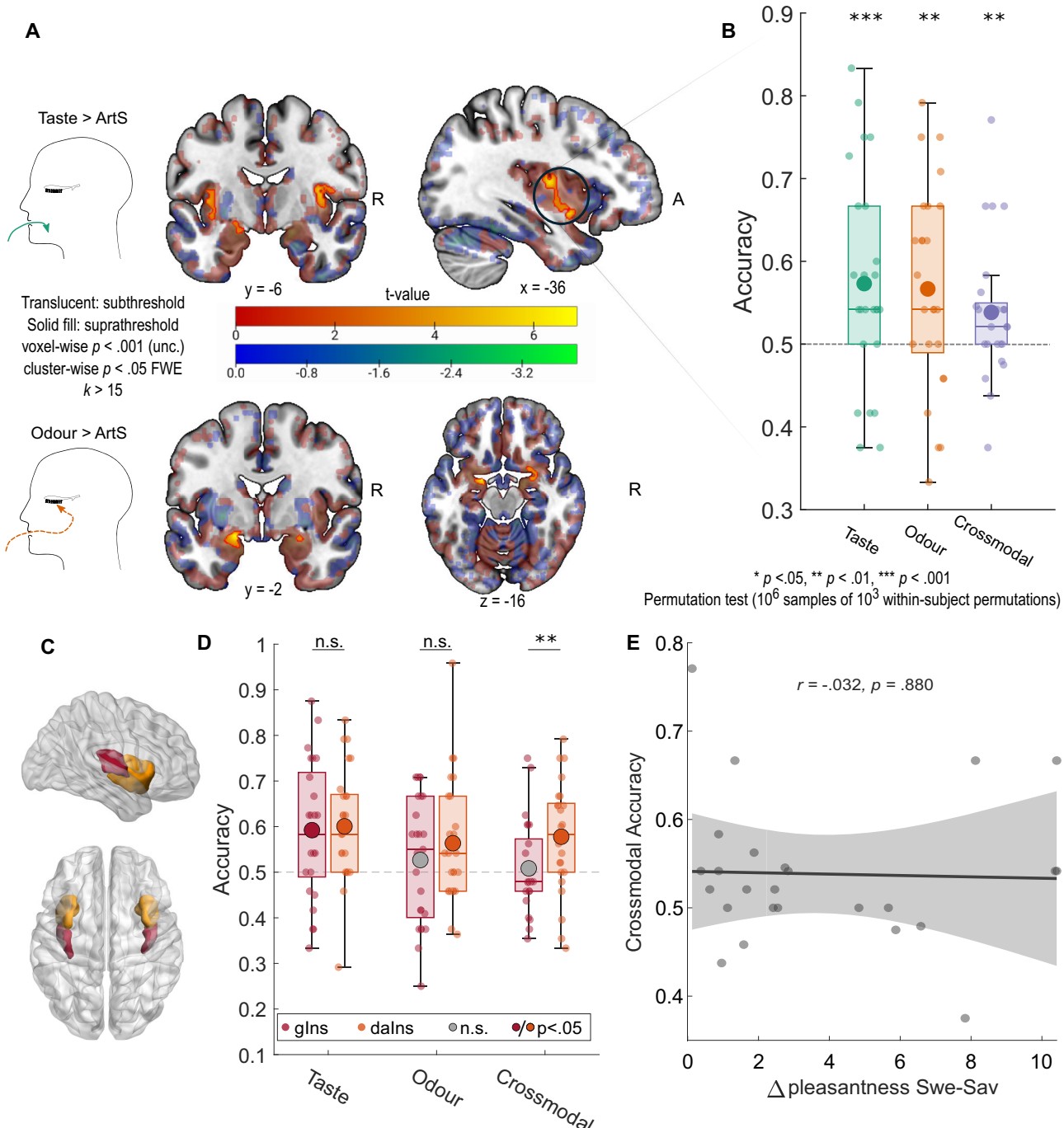

**Fig. 2 | Modality-specific univariate responses and ROI decoding. A** *Top*. Univariate contrast of all taste presentations against tasteless artificial saliva (ArtS) shows BOLD response in bilateral dorsal mid-insula extending to ventral anterior insula. *Bottom*. Univariate contrast of all retronasal odour presentations against ArtS shows a BOLD response in bilateral piriform cortex. **B** Leave-one-subject-out taste ROI decoding analyses showing strong dissociability of taste ($p = 9 \times 10^{-4}$) and odour identity ($p = .003$), as well as crossmodal decoding between taste and odour ($p = .009$; uncorrected comparisons against permuted null distribution). **C** ROIs for granular insula (red) and dysgranular and agranular insula (orange). **D** Significant taste decoding in the dorsal granular insula ($p = 2 \times 10^{-5}$) and significant taste

($p = 9 \times 10^{-4}$), odour ($p = .004$) and crossmodal ($p < 1 \times 10^{-6}$, uncorrected one-tailed comparisons to permuted null distribution) decoding in the dysgranular and agranular insula. Pairwise comparisons between the ROIs were not significant for taste ($p = .760$) or odour ($p = .195$) but significant for the crossmodal decoder ($p = .006$, uncorrected two-tailed permutation-based $t$-tests). Large dots signify means; boxes show the interquartile ranges (IQR); whiskers show $1.5 \times$ IQR. Grey dashed lines signify theoretical chance accuracy. * $p < .05$, ** $p < .01$, *** $p < .001$. **E** Crossmodal decoding accuracy is not driven by hedonic differences between flavours. Shaded areas indicate 95% simultaneous prediction bounds. $N = 25$ participants for all panels. Source data are provided as a Source Data file.

extending from the inferior parietal lobule (IPL) to the bilateral cuneus (peak $z = 2.82$, $p_{TFCE} = .002$), an area that does not typically form part of the flavour network. As each flavour was assigned an abstract visual stimulus, IPL decoding accuracy might be attributed to activity related to the conditioned visual cues. Full results of the searchlight analysis

can be seen in Supplementary Table 4. Supplementary searchlight results of tastant identity decoding can be found in Supplementary Table 5, whereas searchlight results of odour identity decoding did not survive TFCE. Notably, we did not observe above-chance decoder performance for the same partitions in the piriform cortex in either the

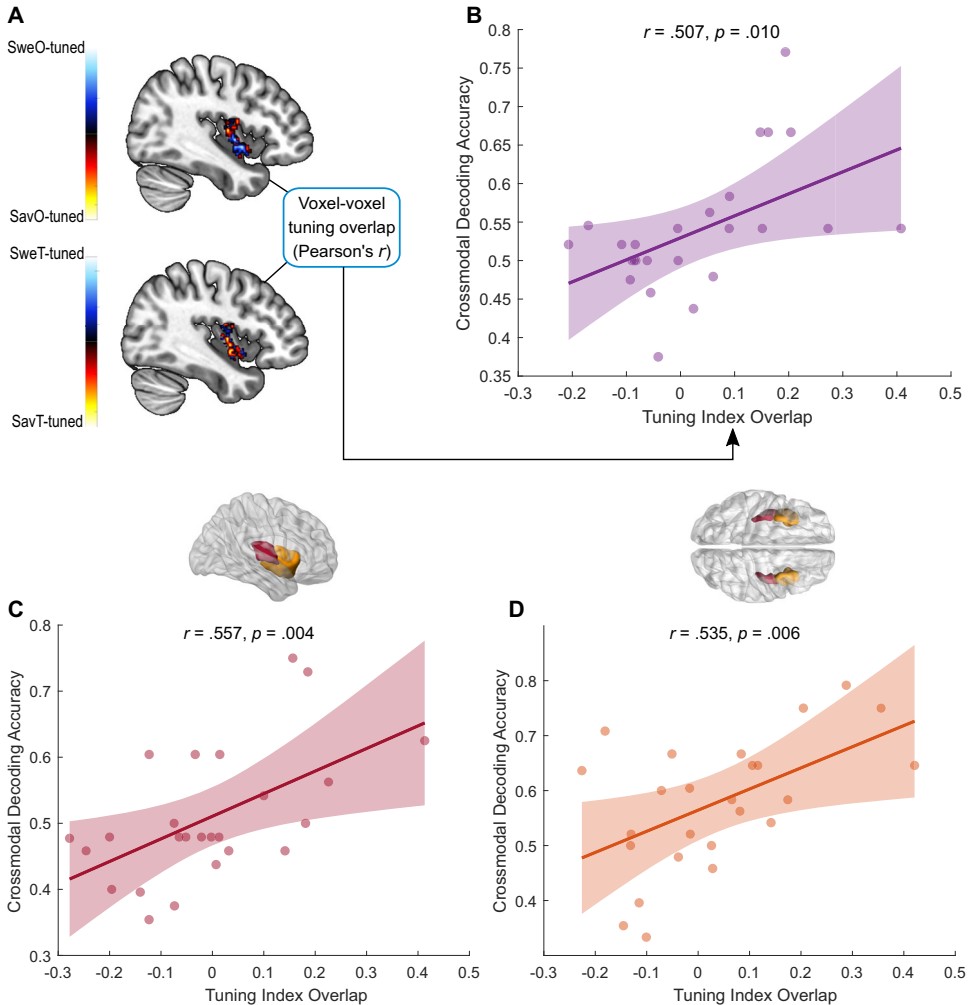

**Fig. 3 | Crossmodal decoding accuracy is driven by the extent of tuning index overlap in all three ROIs. A** Contrasting the statistical parametric maps (T-maps) of sweet against savoury in both modalities results in a spatial map of tuning indices to each condition. Tuning index overlap can be derived by correlating tuning index maps of each modality. **B–D** Tuning index overlap in a given ROI is strongly correlated with crossmodal decoder accuracy in the leave-one-subject-out ROI (uncorrected Pearson correlations for multiple comparisons, $r = .507$, $p = .010$) (**B**), the granular insula ROI ($r = .557$, $p = .004$) (**C**) and the dysgranular and agranular insula ROI ($r = .535$, $p = .006$) (**D**). Shaded areas indicate 95% simultaneous prediction bounds. $N = 25$ participants for panels (**B–D**). Source data are provided as a Source Data file.

whole-brain searchlight analysis or the restricted ROI analysis (Supplementary Fig. 6). Supplementary connectivity analyses indicated that crossmodal accuracy scores in the dysgranular/agranular insula were explained by its effective connectivity to the lOFC, but not by the reverse connection (see "Methods" and Supplementary Fig. 7). Taken together, these results suggest that common representations between flavour components, regardless of modality, exist in various cortical regions beyond established flavour-responsive regions, indicating the presence of a flavour-sensitive cortical network. The fact that the insula is the only canonical primary chemosensory cortex in this network indicates its role as a putative integrating hub.

For cortical regions with modality-independent chemosensory encoding patterns (ventral anterior insula, lateral OFC and IPL), we proceeded to examine differences in how these patterns were organised. A confusion matrix extracted from a spherical ROI of 10 mm around the insula searchlight peak showed that both sweet tastes and 'sweet' odours were more likely to be predicted to be sweet taste, whereas savoury taste and 'savoury' odour were confused for each other (Fig. 4D). Conversely, a confusion matrix extracted from a 10 mm spherical ROI around the lOFC peak shows that stimuli of the same flavour quality (i.e. the odour and taste that were associated with sweetness or savouriness respectively) were

equally likely to be confused by the decoder for each other (Fig. 4E). A similar pattern was also observed for the sweet stimuli in the confusion matrix extracted from a 10 mm spherical ROI around the IPL peak, whereas savoury stimuli of both modalities were more likely to be predicted to be the 'savoury' odour here (Fig. 4F). Ancillary analyses of the confusion matrices can be seen in Supplementary Fig. 8. The differences in these confusion matrices imply encoding differences amongst these regions despite shared flavour-specific patterns in response to flavour components. While sweetness and savouriness are part of different flavour identities, savouriness is rarely encountered as a pure taste in the real world, whereas pure sweet tastants are ubiquitous. For example, while most are familiar with the taste of pure sugar, pure savoury tastes (from sources such as aqueous glutamate) are typically found in mixed savoury dishes that involve multiple sensory properties. We therefore propose that participants were more likely to associate the sweet percept (regardless of modality) with a taste-like sensation, but savouriness as a holistic flavour concept consisting of both aroma and taste. The bias towards predicting sweet stimuli of either modality as sweet taste in the insula, and the lack thereof with savoury stimuli, highlights the putative role of the insula in integrating and encoding the flavour percept.

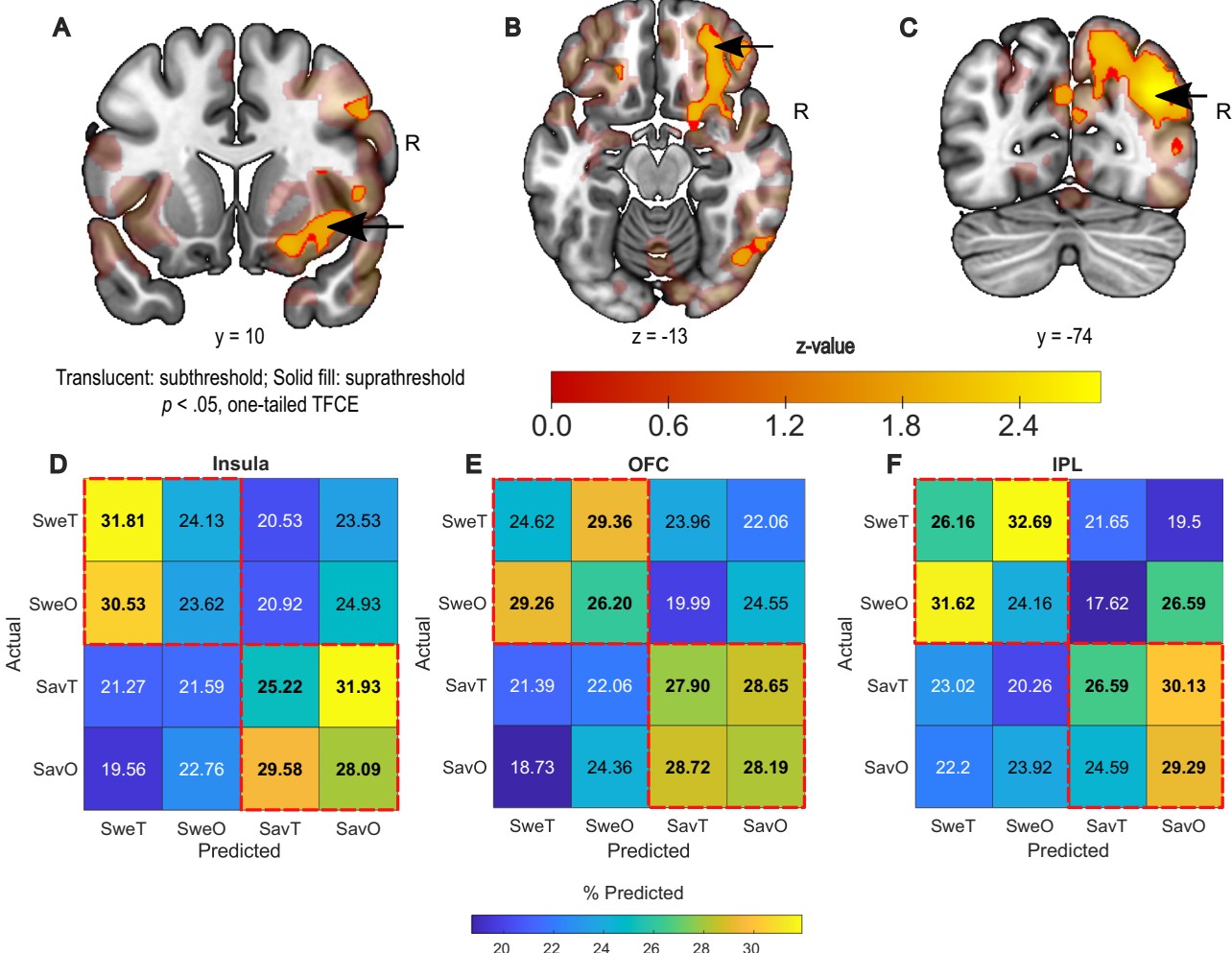

**Fig. 4 | Whole-brain searchlight crossmodal decoding and confusion matrices.**
**A–C** Z-map of searchlight decoding accuracy derived from TFCE, showing significant crossmodal decoding accuracy in the ventral anterior insula (**A**), the lateral orbitofrontal cortex (**B**) and the inferior parietal lobule (**C**). **D** Confusion matrix from the insula peak showing bias for prediction of SweT for both sweet stimuli, whereas prediction for savoury stimuli is not biased towards taste or odour. **E**, **F** Confusion matrix from the OFC (**E**) and IPL (**F**) peaks showing no bias towards sweet taste prediction. SweT – Sweet taste; SweO – Sweet Odour; SavT – Savoury Taste; SavO – Savoury Odour. Red dashed boxes indicate shared flavour identities. Arrows in (**A–C**) indicate the central voxel of the spherical ROIs (10 mm radius) used for (**D–F**). Bolded numbers indicate predictions exceeding the chance level of 25%. *N* = 25 participants for all panels. Source data are provided as a Source Data file.

## Representational drift of flavours over time

While we have shown overlapping taste and odour identity patterns in the insula, we were interested if these patterns change over time, as shown in the rodent gustatory cortex[24,25]. We therefore proceeded to characterise changes in insular taste encoding patterns over scanning sessions and potential similar temporal declines in odour identity representations. Specifically, we compared data trained and tested from the same scanning session against those trained and tested from different sessions in the leave-one-subject-out ROI (Supplementary Fig. 4). Confirming previous observations of temporal change in taste identity representations, decoder performance for taste quality dropped significantly when the decoder was trained and tested on different days (Fig. 5A; LMM β = 0.12, t(92) = 2.05, *p* = .044, 95% CI = [0.0036, 0.2367]), indicating that pattern-based encoding of taste qualities in the insula is not stable across days. This temporal sensitivity effect was not significant for the decoder trained and tested on odour quality or crossmodal decoding (Fig. 5B, C).

While taste encoding specifically appears to show session-related changes, we were also interested in examining if and how flavour representations changed across runs in the same session. To that end, we applied LMM in Representational Similarity Analysis (RSA) using a

quantification of the runwise drift in neural representation. We obtained the scaled pairwise distance between each run number, such that the runs furthest apart had a distance of 1 and proximal runs had the smallest distance (see Fig. 5D). This measure of runwise distance was used as a predictor variable in an LMM in addition to flavour distance to model how well these distances predicted neural dissimilarity between runs (see "Methods"), where distances from the same run were excluded from analysis.

Figure 5D shows baseline representational drift (as judged by the representational drift of the same flavour across runs) in both the insula and the OFC ROIs, defined as a 10 mm radius sphere from the searchlight peaks in these respective regions. There was greater drift in insular flavour representation in the first session compared to the second session (β_{run × day} = 0.020, t(11528) = 2.153, *p* = .0314, 95% CI = [0.0018, 0.0391]) and subthreshold divergence of different flavours (β_{flavour × run} = 0.016, t(11528) = 1.734, *p* = .0829, CI = [−0.0021, 0.0348]). These differences in drift, both the steeper runwise drift on the first day and the trend towards a divergence, are consistent with stimulus identity encoding in the insula, where representations of different flavours are preserved across runs, particularly in the first session. On the other hand, flavour representational drift in the OFC was not

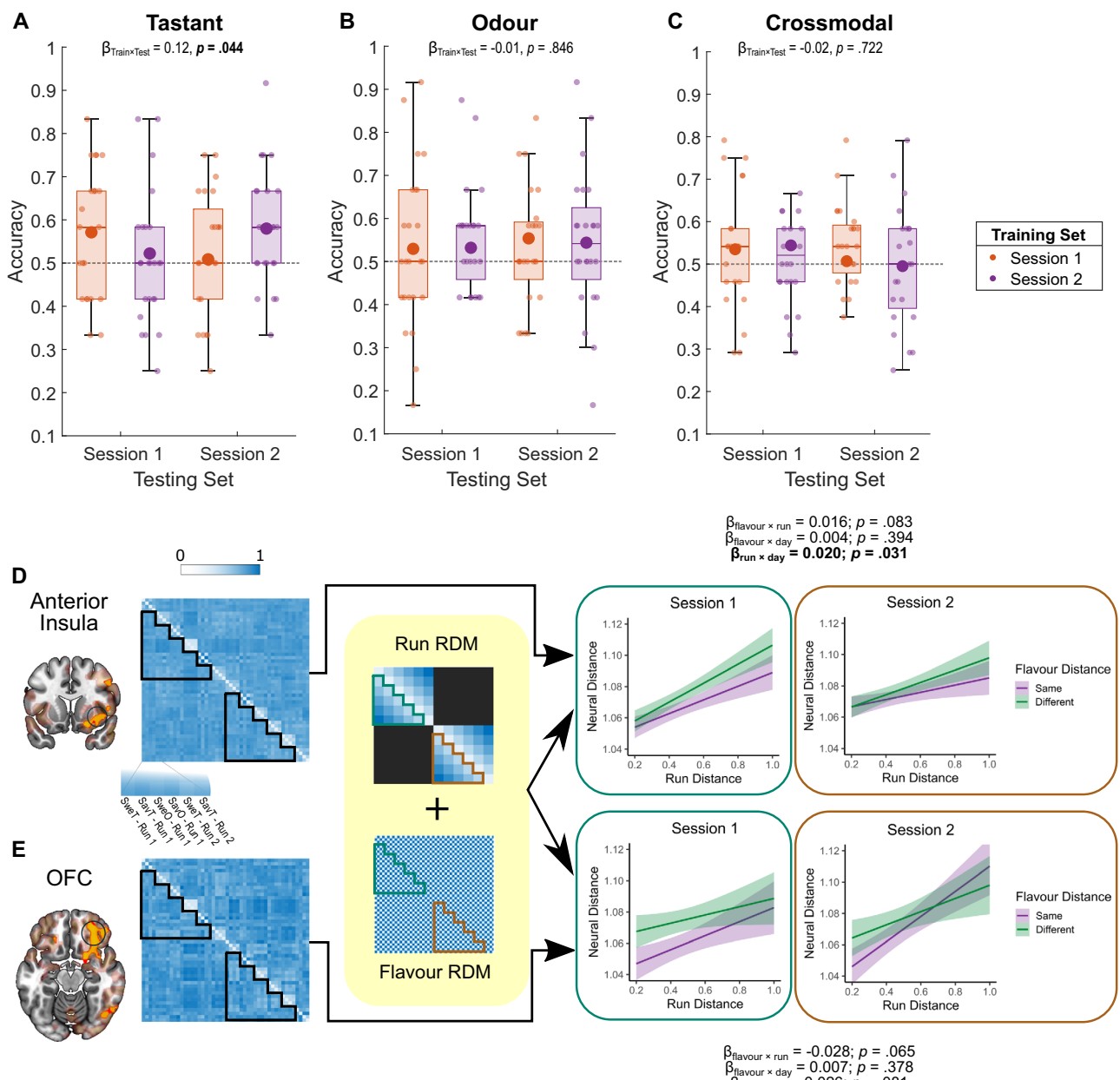

**Fig. 5 | Temporal dynamics of taste, odour and crossmodal neural signals.**
**A** Significant differences in accuracy between decoders trained and tested on taste identity within the same day as opposed to across days. **B**, **C** No significant difference in within-day and across-day accuracies for decoders trained and tested on odour identity or using crossmodal partitioning. $N = 24$. Large dots signify means; boxes show the interquartile ranges (IQR); whiskers show the minimum and maximum ranges using 1.5 IQR. **D** Representational drift of flavour representations in the insula across runs. Model of the neural Representational Dissimilarity Matrix (RDM) from betas of all four conditions across runs (left) as predicted by the runwise distance and the flavour distance. Darker colours represent higher values. Significant reduction in representational drift in the insula between Session 1 and Session 2. **E** Same as (**D**) but using the neural RDM extracted from the orbitofrontal cortex (OFC) searchlight peak, showing no reduction in drift between days. Mixed models included all available data but were not corrected for multiple comparison between partitions and ROIs. Shaded areas show 95% confidence intervals of the linear model. $N = 25$ participants for Session 1; $N = 24$ participants for Session 2. Source data are provided as a Source Data file.

significantly reduced on the second day, showing the opposite trend (Fig. 5E; $\beta_{run \times day} = -0.026$, $t(11528) = -1.751$, $p = .081$, CI = [−0.0570, 0.0032]) and had a subthreshold convergence of flavour representation ($\beta_{flavour \times run} = -0.028$, $t(11528) = -1.847$, $p = .065$, CI = [−0.0580, 0.0017]), indicating that dissimilarity between similar and different flavours was conserved between proximal runs but was less separable between distal runs. This convergence may be indicative of value-level processing in the OFC, as the subjective value of a stimulus is likely to fluctuate over the course of a session, resulting in some preservation of value in proximal runs but a decline in distal runs.

## Discussion

Our study demonstrates that tastes and retronasal odours evoke a shared flavour-specific neural code in the dysgranular and agranular insular cortex, but not in granular insula or piriform cortex. We also developed a quantification of runwise distance to demonstrate that insular representations of flavours change within the course of a scanning session, although this drift is less prominent in the second session. Taken together, these findings provide a neural basis for the quasi-synaesthetic perception of taste and retronasal odour during food consumption. Our findings highlight the role of the insula as a

potential hub for the representation of conceptual overlap between flavour components. This putative early integration explains the strong perceptual link between tastes and odours and can be explained by simultaneous stimulation arising from both modalities during food consumption. The fact that odours and taste evoke comparable neural codes builds the likely neural foundation for the frequently observed confusion between odour and taste during food consumption, as well as their mutual enhancement.

In this study, both sweet and savoury taste stimulation induced activations in the bilateral dorsal mid-insula extending to the ventral anterior insula, which replicates and reinforces previous literature implicating the insula as the primary gustatory cortex[29,30]. In addition to the expected presence of taste quality coding within this taste-responsive region, which was in line with previous MVPA studies[9,31], we also note that this region reliably displays pattern-based encoding of taste-associated odour identity. While the anterior insula and frontal operculum have been known to display activation in response to olfactory cues, particularly food-related odours[20,21], involvement has been less consistently demonstrated for retronasal odour delivery. Here, we demonstrate that, even if the absence of a robust mass-univariate retronasal odour signal in these regions, the human gustatory cortex spatially encodes retronasal odour qualities as well as tastes. These findings echo retronasal odour encoding in the rodent gustatory cortex[14,32], despite differences in gustatory projections between humans and rodents[33,34]. Importantly, our results cannot be explained by a mere overlap in hedonic features, as between-participant variability of hedonic differences did not predict decoder performance. They are also robust to changes in the exact identity of the odours chosen, which varied across participants.

Of notable interest, we show that odour identity patterns in the insula overlap with those elicited by their associated tastes. This provides a putative neural basis for odour-induced taste sensation in the absence of a tastant[3]: that is, a taste sensation occurs because the gustatory cortex responds as it would to a real tastant. This phenomenon has not previously been observed in humans or other mammals, as previous experiments investigating the role of the insula in processing olfactory information have found no response in non-human primates[8,35]; although involvement of the gustatory cortex in odour processing has been shown using human fMRI[20,21,36] as well as rodent electrophysiology[12–14,32,37], our study extends this by showing that retronasal odours share flavour-specific encoding with their associated tastes. While larger effect sizes than those in our ROI have been reported by previous chemosensory decoding studies[9], this ROI was generated from a conservative pre-registered leave-one-subject-out protocol (see "Methods") that prevents information leakage. An anatomically defined ROI (in Fig. 2D) shows a stronger effect. However, a crucial distinction is that, in this study, congruent taste-odour pairings were used, and participants were familiarised with the taste-odour combination to maximise the flavour binding between the chemosensory flavour components. We propose that this strong flavour binding is key, as a taste-odour association is required for an odour-induced taste sensation, which in turn is reflected by taste-like neural patterns in response to odours.

The insula is a large structure with complex variations in cytoarchitecture, most notably the anteroposterior gradient of granularity observed in primates and humans[38–42]. Given the existence of only five known basic taste qualities[43], it is unlikely that a structure with such size and complexity would store basic taste representations without further processing, such as integration with other chemosensory properties of foods. Anatomically, the ventral anterior insula has reciprocal projections from olfactory areas such as the piriform cortex and orbitofrontal cortex[22,23,44], and electrophysiological perturbation of these regions induces neuronal firing in the others[45,46]. In our study, spatial encoding of taste identity, interpreted through the decodability of tastant identity, was spread throughout the granular,

dysgranular and agranular portions of the insula, whereas taste-associated odour encoding was restricted to the dysgranular and agranular insula. This distinction, along with the fact that the insular odour encoding patterns overlap with their associated tastes, implies that the flavour percept is integrated in the dysgranular and agranular insula based on gustatory input from the granular insula and olfactory input from the piriform cortex, as previously proposed by Small et al. (2013)[47]. While inspection of subthreshold data largely indicates symmetrical activation patterns, suprathreshold decoding peaks were predominantly located in the right hemisphere. Given the lack of data on cortical lateralisation in the current participant sample, whether this represents a systematic processing difference remains to be clarified.

The existence of a common neural code for associated gustatory and olfactory information in the insula implies that chemosensory signals that constitute flavour perception are truly integrated into a shared object representation prior to their value-level processing in the OFC. This mechanism goes against the established role of the OFC as the integrating hub of all sensory modalities related to food reward, given direct afferent OFC projections from visual, auditory, tactile, gustatory and olfactory sensory cortices[15]. Nevertheless, early overlapping chemosensory representations in the insula would not necessarily rule out the role of the OFC in evaluating the subjective value of food reward. In humans and other primates, OFC activity does reflect an identity-based value signal in orthonasally presented odours[48–50] as well as nutrient-guided valuation of visual food stimuli[51,52]. In addition to these anticipatory cues, the human and primate OFC is also sensitive to consummatory reward features such as taste[16,36], retronasal odour[36,53] and oral texture[54,55]. Given the established role of OFC neurons in encoding the identity and value of offered and chosen oral food stimuli[17], as well as our finding of cross-modal decoding in the OFC, our results are in line with the idea that the OFC evaluates an integrated flavour signal from the insula.

Building on the evidence of taste-odour integration in the insula, we next examined how these representations change over time. Our study pioneered formal testing of changes in taste encoding patterns across days in humans. Monocellular and ensemble taste identity representations in the rodent gustatory cortex are known to shift[7,24,25]. Furthermore, while such a change has been indicated in a high-field human fMRI study[9], no formal testing of within-day and across-day decoding accuracies was conducted. Inspired by these studies, we designed our experiment to allow testing for similar shifts in humans. In doing so, we show that taste encoding does indeed change across days, with the across-day partitioning strategy performing significantly poorer than the within-day partitioning, thereby indicating that ensemble taste patterns in the human insula also change across days, similar to observations in rodents. By using fMRI RSA to characterise run-wise representational drift of flavour identity in the insula and the OFC, we observed that flavour patterns in the insula drift more on the first day compared to the second day, exhibiting slight divergence. On the other hand, there is pattern convergence and a lack of a day-wise change in representational drift in the OFC. Based on these results, we propose that the insular patterns observed encoded identity, whereas the OFC patterns encoded consummatory subjective value (liking[56]). In the OFC, representational distance between the same flavour and different flavours was largest in proximal runs but similar in distal runs. This implies that the OFC encodes a feature of the stimuli that differentiates flavours but does not remain stable throughout the session. On the other hand, insular representational distance between the same and different flavours is preserved throughout the session, if exhibiting a slight divergence. This implies that the insula holds representations of a stimulus feature that remains constant throughout the session, such as flavour identity.

The low temporal resolution of fMRI does not allow us to directly test that the flavour signal in the insula occurs prior to one in the OFC.

Theoretically, an alternative interpretation might be that taste-like responses in the anterior insula to olfactory stimulation arise due to back-projection after flavour recognition in the OFC[57]. However, our main findings, supplemented by DCM analyses, strongly suggest that retronasal odour representations occur in the insula, combining with functional experiments showing early signalling of retronasal odour in the rodent gustatory insular cortex[12–14,32]. While primate tractography indicates that our results were driven by monosynaptic projections between the piriform cortex and the dysgranular and agranular insula[23,27,44], whether these direct projections or more indirect pathways drove our findings cannot be disentangled using DCM of fMRI signals. Future functional studies in humans should consider methods with higher temporal resolution, such as magneto/electro-encephalography (M/EEG) to answer this question.

In addition to demonstrating overlapping encoding of odours and their associated tastes in the insula, our study expands on previous evidence of the roles of primary sensory cortices in response to unimodal stimuli. Despite robust univariate activation in the piriform cortex in response to olfactory stimulation, our pattern-based analyses were unable to decode between the two odours in the piriform cortex (Supplementary Fig. 6). On the other hand, robust decoding of odour quality was possible in the primary gustatory cortex in the absence of univariate odour activation. Given prior findings demonstrating spatial encoding of odour identity in the piriform cortex for orthonasal odours[6,58], and the established role of the piriform cortex in odour processing in a variety of mammals[6,12,26,32,53], this result was unexpected. One explanation for this divergence may be that the resolution used by our scanning protocol was unable to capture the fine-grained differences in activation patterns between the odour stimuli. While we cannot rule out that decoding would be possible at finer spatial resolutions, the fact that odour identity decoding was in fact possible in the insular ROI may also indicate fundamental differences in identity coding for orthonasally and retronasally presented odours. Therefore, future research should consider higher spatial resolutions to resolve this issue.

For the present study, avoiding trial-by-trial ratings, particularly of pleasantness, was a conscious decision based on previous evidence showing task-based variability in insular activation[59]. Tasks involving explicit pleasantness ratings inflate differential activation between taste-containing and tasteless stimuli compared to detection, identification and passive tasting paradigms[60]. Electing for a passive tasting paradigm, similar to Avery et al. (2020)[9], ensured that the observed taste signal was driven by the presence of a tastant, rather than attentional effects. Furthermore, we combined explicit pleasantness ratings collected on both days with correlation methods in supplementary analyses (Fig. 2E), where we do not detect a relationship between differences between hedonic ratings and crossmodal accuracy, thus indicating that the difference in pattern-based signals is not driven by changes in pleasantness. Due to the lack of trial-by-trial psychophysical and hedonic ratings, we are, however, unable to authoritatively distinguish between cortical sites where overlapping encoding of food odour and taste is consistent with identity-based versus value-based encoding. An avenue for future research is to develop a design that would allow the dissociation of these possibilities in a targeted manner.

One natural extension of this work is to investigate potential crossmodal encoding between orthonasal olfaction and gustation. Odour is the only food sensory property that is perceived in the anticipatory and consummatory phase, due to orthonasal and retronasal routes of olfaction[3]. Given the role of odour in encouraging approach or avoidance[61], crossmodal encoding of gustatory and retronasal olfactory cues in the insula may form part of a mechanism whereby odours acquire taste properties during consumption to modulate appetitive behaviour towards sources of the same odour in the outside world[62]. This mechanism, however, hinges on a common representation of odours and tastes of the same food item, particularly

in cortical regions that respond to gustatory information. Rodent gustatory cortex ensemble patterns for retronasal odours do not overlap with tastes[14], and inactivation of the gustatory cortex inhibits retronasal olfactory learning while maintaining orthonasal olfactory learning[37]. However, rodents are obligate nose breathers[63], whereas humans can and do breathe through the oral cavity[64], such that there may be more similarity between retronasal and orthonasal olfaction in humans. Furthermore, although single-neuron recordings in the primate insula show no encoding of orthonasal stimulation[8,35], odours forming part of a flavour percept through constant contingent presentation with a taste stimulus may induce a gustatory cortical response. Moreover, due to the ensemble nature of taste encoding[43], single-neuronal recordings may fail to capture taste-related ensemble activity. Our work also provides a theoretical foundation to further explore the mechanisms of odour-induced taste enhancement, which arise when a conceptually matching odorant and tastant are presented concurrently. Our data indicate a potential functional mechanism by which this mutual enhancement could be implemented, but testing it on multisensory stimulus combinations would have introduced confounds and was beyond the scope of this study.

Taken together, our study demonstrates the role of the insula as a critical hub for flavour integration through taste-odour convergence. Our crucial finding on the crossmodal overlap between odour encoding patterns and their associated tastes extends previous functional neuroimaging studies in various mammalian species implicating the insula as the primary gustatory cortex[8,10,11,35,65], and evidence for reciprocal insular-piriform projections established through tractography[23,27,44] and neuronography[45,46]. This finding both explains observations in behavioural studies of odour-induced taste sensations[3] and likely forms part of a larger mechanism whereby odours acquire taste properties and associated hedonic values to encourage or discourage more consumption of particular foods. This basic mechanism behind flavour integration has implications for flavour preference acquisition and dietary patterns.

## Methods
### Participants

Twenty-eight participants in total completed the fMRI sessions. Of these, 3 had to be excluded due to excessive signal dropout (1), excessive movement during scanning (1) and nausea (1). Therefore, 25 participants (11 male, 14 female, by self-report) were used in the analysis, of whom one attended only one scanning session. Behavioural ratings from the first session of one participant were lost due to a clerical error, such that between-day analyses of ratings only included 23 participants. The mean age of the sample was 27.8 years (SD = 6.0 years), and the mean BMI was 22.8 kg/m$^2$ (SD = 2.8 kg/m$^2$). To be included, participants had to be between 18 and 45 years old, speak fluent English, have a normal sense of smell (tested using the Sniffin' Sticks Identification task [cut-off score of 12][66,67]) and a normal sense of taste (tested using tastant sprays in the mouth), have normal or corrected-to-normal vision, not be pregnant, display no cold/flu symptoms and have no known eating disorder. All participants completed an informed consent sheet before taking part in any screening. All procedures were in accordance with the Declaration of Helsinki and approved by the local ethics committee (Regionala etik-prövningsnämnden i Stockholm, Dnr 2021-05138). Participants were given 600 Swedish kronor in gift cards as compensation upon completion of all parts of the study.

### Laboratory session

Participants were trained on the task and the stimuli pairing simultaneously in their pre-scanning behavioural session. This session served multiple purposes:

- It allowed the selection of the specific odorants used for each participant, ensuring approximate isointensity.

- It ensured that our results were generalisable, as we selected different odorant combinations (see Supplementary Fig. 2A).
- We trained participants on taste-odour pairings by presenting them taste-odour combinations in repetition, as taste-odour associations can be culture-specific, and we recruited from an international population of English speakers in Stockholm.
- Participants learned the abstract symbols representing sweet and savoury flavours, ensuring that we could test stimulus identification without relying on potential biases arising from explicit semantic labels such as 'sweet' or 'savoury'.
- Participants were trained to consume small aliquots of liquid (1 ml), delivered via a similar setup as in the scanner, while in a supine position on a massage bench, which minimised discomfort and the potential for choking during the actual scan.

At the start of the training session, participants filled out a screening questionnaire and performed the taste and odour screening, during which they identified taste qualities of tastants sprayed in the mouth and the odour identities of common objects using the Sniffin' Sticks protocol[66,67]. Subsequently, each participant completed a short pretesting task by rating the pleasantness and intensity (both on a visual analogue scale between −5 and 5) of three sweet and three savoury flavour mixtures three times in a randomised order from taste cups. Two flavour mixtures, one sweet and one savoury, matching in mean intensity were chosen for the study. Mixtures with a pleasantness rating below −2 were not considered in order to prevent evoking disgust responses in the participant. These mixtures were subsequently used during a series of identification and rating tasks for the rest of the behavioural session. Therefore, each participant had a bespoke sweet and savoury flavour combination.

Prior to the session, participants were randomly assigned two abstract visual cues (letters from the Phoenician alphabet), one for the savoury flavour and another for the sweet flavour. In each trial of the rating task, the participant was presented 1 ml of a stimulus and was asked to choose which visual cue it corresponded to, such that participants would learn the cue associated with each flavour. During this task, the word 'Identify' was presented at the top of the screen, and participants could choose between the two visual cues or decline to answer, which would lead to no response. This cue association allowed us to examine if participants were able to distinguish the flavours without being primed by words such as 'sweet' or 'savoury'. They subsequently rated it for pleasantness and intensity, during which one of the words 'Pleasantness' or 'Intensity' was presented at the top of the screen. Participants responded by moving a cursor on a visual analogue scale. For the pleasantness task, the scale was anchored at 'Extremely disgusting' at the bottom, 'Neither' in the middle, and 'Extremely pleasant' at the top. For the intensity task, the scale was anchored at 'Not at all intense' at the bottom and 'Extremely intense' at the top. Parts of the trial that required participant input ('Identify', 'Pleasantness' and 'Intensity') lasted for a maximum of 5 s but ended after the participant entered the input. Therefore, each trial ranged between 10 s to 25 s. Participants completed up to 60 trials, although from trial 40 onwards, the experiment ended once their cumulative accuracy rate for correctly identifying a given flavour stimulus with the correct visual stimulus was above 85%, thereby ensuring that the participants could sufficiently distinguish between the two flavour stimuli.

## MRI sessions

Participants were instructed to attend two structural and functional MR sessions, scheduled such that the final MR session was within 10 days of the behavioural session. During the MRI sessions, participants orally received unimodal stimuli—that is, only sweet taste (SweT), savoury taste (SavT), sweet odour (SweO) or savoury odour (SavO) during the mini-block, as opposed to a flavour combination, in addition to artificial saliva. Each participant had bespoke SweO and SavO stimuli derived from the flavour combinations used in their behavioural laboratory session. Liquid stimuli were orally delivered using custom 3D-printed mouthpieces designed for gustatory stimulation in fMRI contexts[68] connected to peristaltic pumps.

Each MRI session began with a shortened version of the behavioural task from the laboratory session (24 trials), where they performed the identification and rating tasks in the MR scanner bore, prior to any scanning, using the same visual cues that they were assigned in the laboratory session. Participants were briefed that the flavours they would receive might vary slightly from the ones in the laboratory session, such as in terms of intensity. Only participants who scored over 75% accuracy could continue (for full breakdown, see Supplementary Fig. 3). All participants achieved this score or higher. Between the rating task and the functional runs, participants also underwent a structural multi-echo MPRAGE scan to facilitate image realignment and normalisation.

Participants completed six functional runs per session. Participants were presented with 0.5 ml of the unimodal stimuli or artificial saliva and asked to swallow. This sequence was repeated four times with the same stimulus to form a mini-block (for a total of 2 ml over 16 s) and followed by a rinse block consisting of 1 ml of artificial saliva before moving to the inter-trial interval. Between mini-blocks, participants were presented with a grey fixation cross for 8–12 s. Participants were instructed to swallow only when the swallow cue appeared on the screen. The stimulus order was pseudo-randomised such that every stimulus (including artificial saliva) has 3 repetitions per run and that stimuli were not repeated consecutively more than once, and each run consisted of 15 mini-blocks.

## Stimuli

**Unimodal stimuli.** The sweet taste (SweT) stimulus was 9% w/v sucrose (Sigma Aldrich). The savoury taste (SavT) stimulus was 1% monosodium glutamate (Sigma Aldrich).

Sweet-associated odorants (SweO) used were:
- Golden syrup (DSM-Firmenich)
- Lychee (DSM-Firmenich)
- Raspberry

Savoury-associated odorants (SavO) used were:
- Smoky bacon (DSM-Firmenich)
- Chicken (DSM-Firmenich)
- Onion (DSM-Firmenich)

All stimuli were dissolved in artificial saliva (ArtS; 25 mM KCl + 2.5 mM NaHCO$_3$ in distilled water). In addition, ArtS was used as a baseline stimulus and rinse solution in the functional MR task in order to present a baseline condition of oral stimulation without activation of the taste cortex[69].

During piloting, pure odorants dissolved in ArtS were tested by nose-clipped volunteers and asked for verbal descriptions of taste sensation, whereupon volunteers attested to their tastelessness. Additional triangle testing of the odorant solution was performed to ensure tastelessness. Three taste cups containing ArtS were presented to nose-clipped volunteers, where two contained only ArtS and one the odorant dissolved in ArtS. This process was repeated for eight trials, such that a binomial test could be carried out, where six of eight correct identifications would yield a significant one-tailed binomial test at α < .05. This technique has previously been used to demonstrate lack of discriminability[70]. In total, five volunteers performed the task, where four volunteers tested four odorants, and one volunteer tested two odorants. Therefore, each odorant was tested by three volunteers.

Supplementary Fig. 1 shows that the nose-clipped volunteers performed at around chance level, and no volunteer could significantly distinguish the odorant from plain ArtS solution.

Each participant completed a short pre-testing task before performing the identification and rating task in the laboratory session. The unimodal odorants used on MRI days were the same as the odorants used in the flavour stimuli in the laboratory sessions. For example, a participant who had sucrose + lychee and MSG + chicken in the laboratory session would have sucrose, MSG, lychee and chicken separately in the MRI sessions.

**Flavour stimuli.** Flavour stimuli were created by combining the unimodal stimuli. In general, tastants (either sucrose or MSG) were added to the unimodal odour stimuli to achieve the same concentration as the unimodal stimuli; that is, the flavour stimuli had the same concentration of the odorant as the odour stimuli and the same concentration of the tastant as the taste stimuli.

**Visual.** Each flavour pairing was associated with a visual stimulus cue to avoid 'sweet' and 'savoury' semantic labels. The visual stimuli (characters from the Phoenician alphabet) were processed, and 14 were deemed suitable for this purpose. Participants were randomly assigned stimuli (one for the sweet flavour and one for the savoury flavour) before training.

### MRI data acquisition and preprocessing
**Acquisition.** Each run consisted of 228 T2*-weighted BOLD gradient multi-echo echoplanar images (EPI) using a Siemens 3 T Prisma scanner running the syngo MR E11 system equipped with a 64-channel head coil. Fifty-two interleaved axial slices of 1.7 mm thickness were collected with the following parameters: in-plane voxel size = 2 mm × 2 mm; slice thickness = 1.7 mm (distance factor of 15%); echo time (TE) = 42.0 ms; repetition time (TR) = 2000 ms; flip angle (FA) = 90°; field of view (FOV) = 208 mm × 208 mm. For brains that exceeded the field of view, partial brain coverage was used, which included the frontal lobe, temporal lobe and occipital lobe.

Prior to the functional scans, a gradient echo image with two echoes was extracted to generate a field map with the following parameters: 2 mm isotropic voxel; TE1 = 4.92 ms; TE2 = 7.38 ms; FA = 60°; TR = 565 ms; FOV = 208 mm × 208 mm; phase encoding direction: R → L axial.

Furthermore, to enable normalisation to a standardised space, a high-resolution multi-echo T1-weighted MPRAGE structural image was acquired with the following parameters: 1 mm isotropic resolution sagittal slices; TE1 = 1.69 ms; TE2 = 3.55 ms; TE3 = 5.41 ms; TE4 = 7.27 ms; FA = 7.0°; acquisition time = 2530 s; FOV = 256 mm × 256 mm. Extracting the root-mean-square (RMS) of the multiple echoes resulted in a high-quality image that was used in subsequent steps.

**Preprocessing.** Images were preprocessed using SPM12 (Wellcome Department of Imaging Neuroscience, Institute of Neurology, London, UK) implemented in MATLAB 2021b (Mathworks). Firstly, a field map for each MRI session was calculated using the gradient echo images, specifically the phase images and the magnitude image, using the in-built SPM12 Field Map Toolbox. Random EPI images from the session were then loaded and unwarped to visually check the extent of distortion correction. The structural images from both MRI sessions were coregistered, and a mean image was extracted in order to further improve spatial resolution. This mean structural image was then segmented into tissue probability maps (TPMs), and a skull-stripped brain consisting of the grey matter, white matter and cerebrospinal fluid was calculated (with a combined TPM cutoff of 0.8).

After slice-time interpolation to the 0.99-second slice, functional images from each run were realigned to the first image and unwarped using the previously calculated field map for the pertinent session.

They were then coregistered to the skullstripped brain and normalised to MNI space at 2 mm isotropic resolution using the deformation field obtained from the segmentation phase. The resultant normalised images were then smoothed with a 6 mm full-width-half-maximum (FWHM) isotropic Gaussian kernel. Finally, both the smoothed and unsmoothed normalised images from each run were detrended using Linear Model of Global Signal detrending[71] to remove global effects from the time series on top of the pre-registered preprocessing pipeline. These detrended images were used in further analyses.

### Statistical analyses
**Mass-univariate GLM analyses.** First-level general linear models (GLMs) were conducted within each participant using the smoothed and normalised individual BOLD data in SPM12. Within each individual subject, we modelled the following regressors of interest (using a boxcar function with a duration of 16 s) for each run:

- All tastant presentations (both sweet and savoury – Condition 1)
- All odorant presentations (both sweet and savoury – Condition 2)
- All ArtS presentations as the explicit baseline condition (Condition 3)

We also modelled the rinse block for a period of 4 s as a regressor of no interest. The boxcar regressors were convolved with the canonical haemodynamic response basis function (HRF). Furthermore, confound regressors included the six rigid-body motion parameters estimated from the realignment procedure, in addition to spike regressors to censor frames with framewise displacement greater than 1 mm (as calculated from the motion parameters). Runs with more than 10% of their frames censored were excluded from the analysis. A grey-matter explicit mask was used (SPM grey matter TPM thresholded at 0.2) to limit the analysis to only voxels containing grey matter. We also specified the model to include a high-pass filter (HPF) of 128 s to remove slow signal drift. The design matrix included a runwise intercept and applied the default SPM grand mean scaling, such that the mean over space and time within a run was scaled to 100.

Within each subject, a contrast of interest (Condition 1−Condition 3 in the case of tastants and Condition 2−Condition 3 in the case of odorants) was calculated. The group-level analysis using a one-sample t-test on the globally pooled contrast of interest using an unweighted summary statistics approach. Analyses were conducted using a whole-brain grey-matter mask. Cluster-wise significance was conducted using a threshold of $P_{FWE} < .05$, a cluster-cutting threshold of $k > 15$ and a voxel-wise threshold of uncorrected $P < .001$. Small-volume corrections (SVCs) were performed from pre-registered peaks, limiting them to a 10 mm radius spherical Region of Interest (ROI). Piriform SVC coordinates were taken from a statistical localisation of the olfactory cortex[26], whereas anterior insula SVC coordinates were taken from a high-field study on taste activation[9]. ROIs for small-volume corrections and the analysis steps were pre-registered, although we applied a stricter cluster-cutting threshold than pre-registered.

**Multivariate pattern analysis.** Multivariate Pattern Analysis (MVPA) was conducted using the CoSMoMVPA toolbox implemented in MATLAB 2021b. For ROI analyses, the ROIs were formed from functional clusters of the contrast of tastants against ArtS in the mass-univariate GLM analysis to isolate regions that were responsive to taste. In order to avoid bias by using the same participant for ROI generation and analysis, ROIs for each subject were generated from the data of the remaining subjects (leave-one-subject-out cross-validation), using a more lenient voxel-wise threshold of uncorrected $P < .01$ and a cluster-cutting threshold of $k > 150$. Post-hoc ROI MVPA used subregions of the insula as parcellated in the Brainnetome Atlas by Fan et al. (2016), which uses a combination of probabilistic diffusion tractography, connectivity-based parcellation, cross-validated optimal clustering, anatomical connectivity patterns and resting-state

functional connectivity patterns from a subset of the Human Connectome Project data. To limit idiosyncratic differences in myelocytoarchitecture, atlas subregions 163/164 and 171/172 were merged into a granular ROI, whereas the rest of the insula formed the dysgranular/agranular ROI.

Prior to any MVPA, first-level GLM analyses were conducted on unsmoothed normalised functional data in order to preserve voxel-level differences in activation. Within each subject, we used the following regressors of interest (using a boxcar function with a duration of 16 s) for each run:

- Sweet taste presentation (SweT)
- Savoury taste presentation (SavT)
- Sweet odour presentation (SweO)
- Savoury odour presentation (SavO)
- Artificial saliva explicit baseline presentation (ArtS)

In addition, rinse periods were modelled as regressors of no interest for a period of 4 s. Regressors were convolved with the canonical HRF, and confound regressors were the same as the univariate analyses. Similar to the univariate analyses, we also included an HPF of 128 s and applied a whole-brain grey-matter mask. The design matrix included a runwise intercept and applied the default SPM grand mean scaling, such that the mean over space and time within a run were scaled to 100.

MVPA was conducted on the resultant runwise betas of the above model. As pre-registered, we trained a linear support vector machine (SVM; implemented in the LIBSVM package[72]) decoder on the beta weights for each voxel obtained from the first-level GLM. We employed a leave-one-run-out cross-validation partition: in each decoding step, the decoder is trained on all runs bar one, and its performance is subsequently tested on the left-out run. For crossmodal and across-session decoding analyses, we subtracted the training data mean from both the training and testing data to remove mass univariate differences in activation between different modalities. Whole-brain crossmodal searchlight MVPA used the unsmoothed detrended functional scans in MNI space. At each spherical searchlight with a radius of 3 voxels (6 mm), we applied the same partitioning and mean-centring strategy as the ROI analysis to train and test the decoder. Only grey-matter voxels were used to create the searchlights. We then mapped the average accuracy of the decoder onto the centre voxel of the searchlight before moving on to the next searchlight. We subtracted the theoretical chance level from the accuracy maps before then smoothing them using a 6 mm FWHM isotropic Gaussian kernel for group-level analysis. The searchlight radius used was smaller than pre-registered (9 mm) to improve the specificity of the searchlight analyses.

**Tuning index overlap comparison**

In order to show that crossmodal decoding accuracies of all three regions were indeed reflective of similar or overlapping patterns of activations across modalities, we compared the tuning indices of each voxel in the various ROIs. Specifically, we took the statistical parameter maps (T-maps) of each condition of interest (SweT, SavT, SweO and SavO) and created T-contrasts for each modality by subtracting the respective sweet T-maps from savoury T-maps. That is, tastant tuning index maps were created by subtracting the SweT map from the SavT map, and odorant tuning index maps were created by subtracting the SweO map from the SavO map (Fig. 3A). Therefore, modality-specific univariate activations were removed at the voxel-level, and positive tuning indices reflected the extent to which each voxel was tuned to the sweet stimuli (SweT or SweO, depending on the modality) against the savoury stimuli (SavT or SavO). We then used Pearson's correlation coefficient ($r$) to examine the extent to which tuning indices overlapped between modalities. We subsequently correlated this tuning index overlap against the crossmodal decoding accuracy scores of

each ROI to examine the extent to which accuracy scores could be predicted by overlapping tuning indices.

**Effective connectivity analysis**

We used Dynamic Causal Modelling (DCM) to characterise associations between crossmodal decoding accuracy in the dysgranular/agranular insula (daIns) and directed information flow in an anatomically defined network of ROIs, namely the granular insula (gIns), daIns, piriform cortex (Pir) and lOFC. A fully connected DCM was constructed with driving inputs to the gIns for taste stimuli, Pir for the odour stimuli and both gIns and Pir for the ArtS condition. In addition, the DCM had a full intrinsic, steady-state and bidirectional connectivity matrix (A matrix) between all four regions. We did not specify modulatory effects on the connections between regions. After Volume of Interest (VOI) extraction from each specified region in each participant (concatenating the design matrix across runs with run-specific intercepts), we estimated the full DCM for each participant. We then used a second-level parametric empirical Bayes approach (PEB) to prune the model to a set of parameters that best explain the data[73], with a constant intercept and each participant's daIns crossmodal accuracy in the design matrix. This results in a Bayesian hierarchical model with estimates for each connection specified in the A matrix, as well as the posterior probabilities derived by comparing the evidence for all models in which the specific connection was switched on versus all models where it was switched off[74]. Parameters were estimated for the commonalities (average connectivity across participants, orthogonal to crossmodal decoding accuracy in daIns) as well as the connections associated with crossmodal decoding accuracy in daIns. We displayed connections thresholded at a posterior probability (Pp) > .95 for the corresponding parameter.

**Permutation-based significance testing**

Significance testing of classification accuracies was performed via nonparametric permutation testing instead of the pre-registered parametric $t$-tests against the hypothetical chance level. We elected to perform permutation testing to establish a data-driven null distribution that would account for potential autocorrelations or biases that might lead to differences between a theoretical and a data-driven null distribution[75,76]. Specifically, in ROI MVPA analyses, the null distributions were created through $10^6$ samples of $10^3$ within-subject permutations. That is, we shuffled the labels (SweT, SavT, SweO, SavO) within each subject and trained and tested the decoder following the same partitioning parameters $10^3$ times. From these within-subject null distributions, we then sampled one accuracy per subject and obtained the group mean accuracy $10^6$ times with replacement. We then calculated one-tailed $P$-values based on the ratio of permuted group mean accuracies higher than the real group mean accuracy. Pairwise comparisons of accuracies in the granular insula against the dysgranular and agranular insula were performed using permutation-based paired $t$-tests, implemented in the PERMUTOOLS toolbox[77], and effect sizes were derived using the repeated-measures Cohen's $d_{rm}$ method[78].

For whole-brain searchlight analyses, the theoretical chance level (one divided by the number of labels used in the testing set) was subtracted from the accuracy maps from searchlight analyses. These maps were then smoothed and masked such that only grey-matter voxels were used. Finally, we performed prevalence testing on the group data using $10^4$ Monte-Carlo simulations and threshold-free cluster enhancement to account for multiple comparisons[79] implemented in CoSMoMVPA. Clusters of more than 15 contiguous voxels with a z-score of greater than 1.65 (for one-tailed testing) survived.

**Representational drift analysis**

In order to analyse flavour representational drift across runs, we employed a combination of representational similarity analysis with linear mixed-effects modelling and a measure of runwise distance we

developed. Specifically, we quantified the runwise distance as the pairwise distance between runs in the same session while excluding data from separate sessions (Fig. 5D, E), scaled by dividing the distances with the largest runwise distance. Parallel to this, we extracted the betas for each of the four conditions in each run before creating a representational dissimilarity matrix (RDM) by extracting the pairwise correlational distance between them (defined as one minus the voxel-voxel Pearson correlation coefficient). We then designed a linear mixed-effects model with where the neural RDM is predicted by the runwise RDM, the flavour RDM (0 for the same flavour; 1 for different flavours) and the session indicator, as well as their interactions, with the subject identity as the random effect and random intercepts and random slopes for the runwise RDM, the modality RDM, the flavour RDM and the session indicator. Linear mixed-effects models were conducted using the inbuilt fitlme function on MATLAB 2021b with the following formula:

$$
\begin{aligned}
neural\ RDM \sim\ &Flavour\ RDM \times Run\ RDM + Flavour\ RDM \\
&\times Session + Modality\ RDM + Session \times Run\ RDM \\
&+ (Flavour\ RDM + Modality\ RDM + Session \\
&+ Run\ RDM | Participant\ ID)
\end{aligned}
$$

## Visualisation
Box charts and scatter plots were created using MATLAB 2021b. T-maps and Z-maps were visualised using MRIcroGL (https://www.nitrc.org/projects/mricrogl/). ROIs were visualised using the BrainNet viewer (https://www.nitrc.org/projects/bnv/)[80].

## Pre-registration
The design, preprocessing and univariate and taste-responsive ROI MVPA decoding (including across-day MVPA) analyses were pre-registered, along with ROIs (https://osf.io/a8mte). MVPA on the parcellated insular ROI and representational drift analysis across runs were exploratory.

## Reporting summary
Further information on research design is available in the Nature Portfolio Reporting Summary linked to this article.

## Data availability
The first-level betas generated in this study have been deposited in the Zenodo database [DOI: 10.5281/zenodo.16875496; https://zenodo.org/records/16875496]. Access can be obtained by filling out a guest access request. Unthresholded group-level data is available on https://neurovault.org/collections/OOBZESPX/. Source data are provided with this paper.

## Code availability
MATLAB code is available on OSF (DOI: 10.17605/OSF.IO/2KRYV).

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

## Acknowledgements

This project has received funding from the European Research Council (ERC) under the European Union's Horizon 2020 research and innovation programme (grant agreement n° 947886) to JS and by the Swedish Research Council (VR 2018-0318 and VR 2022-02239) to JS. The authors would like to thank the staff at the Stockholm University Brain Imaging Centre for their help and support in preparing the scanning protocols; DSM-Firmenich and Prof. Thomas Hummel for providing the tasteless aromas used in the study; Dr Christoph Pfeiffer for help with the 3D-printing of the mouthpiece; Zahra Hejazi, Sümeyra Nur Doğan, Hanne Helming and Hilda Lindén for scanning and lab assistance; Dr Gustav Nilsonne for help with pre-registration and Open Science efforts; Leonie Seidel, Dr Anna Gerlicher and members of the Perception Lab for helpful comments and discussions. Data acquisition was supported by a grant to the Stockholm University Brain Imaging Centre (SU FV-5.1.2–1035-15).

## Author contributions

P.A.K. and J.S. designed the experiments. P.A.K. performed the research. P.A.K. and M.G.V. conducted the analyses. J.S. supervised the research. P.A.K. and J.S. wrote the manuscript. All authors substantially revised and edited the manuscript.

## Funding

## Competing interests

The authors declare no competing interests.
