## [Transparent Peer Review file · Nature Communications]

Tastes and retronasal odours evoke a shared flavour-specific neural code in the human insula

Corresponding Author: Dr Putu Khorisantono

Version 0:

Reviewer comments:

Reviewer #1

(Remarks to the Author)

The work in this manuscript addresses a relatively simple question: if certain smells are described as “sweet” (or “savory”), will retronasal delivery of those smells activate gustatory cortex, and will that activation resemble the activation driven by (congruent) sweet taste delivery? The authors’ analyses of fMRI BOLD signals provide evidence supporting this hypothesis. They go on to provide evidence that: 1) this phenomenon is restricted to ventral gustatory insular cortex, and not to dorsal insular or olfactory piriform cortex; and that 2) the taste-smell similarity fades in comparisons made between sessions, which they interpret to mean that taste responses “drift” (more than olfactory responses).

There’s a lot to like about this brief report, and I am enthusiastic about seeing them added to the literature. The conclusions are nontrivial (general-language descriptors like “sweet” don’t necessarily refer to an actual taste; when we call a person “sweet,” we are seldom imagining a taste sensation), and for the most part (see minor quibbles below) the analyses seem rigorous and well done. I am particularly pleased with the way that the authors pre-specified specific hypotheses, and the careful, elegant use of the searchlight. There are only a few issues that I’d like to see addressed before I sign off on publication.

Central suggestions.

1) The authors repeatedly make an error in interpretation of their statistics that has been the topic of much discussion (see, for example, Nieuwenhuis, S. et al., *Nat Neurosci*, 14), wherein they perform separate analyses of two datasets, but then interpret their data as revealing significant differences BETWEEN the datasets. This is illegitimate: a significant effect found in only one of two datasets does not mean that the two datasets differ from one another; for instance, a $p = 0.04$ on one dataset and a $p = 0.06$ on the other will likely translate into a $p \sim 0.9$ with regard to the difference in the two effects.

The first of these errors has to do with Figures 2C-E. For the authors to convincingly reach the conclusion that they want to reach with these panels—which is that ventral insular codes odors and x-modality pairs BETTER than dorsal insular—they must do a direct comparison of the results for the 2 regions and show a difference. A multivariate ANOVA showing a significant region x stim-type interaction, followed by post-hoc comparisons, would do the job. Even better would be planned comparisons, which are (less conservative and) justified by References 23 and 27.

Similarly, the authors wish to convince the reader that odor coding is more stable across days than taste coding (Figures 4A-C). In this case they are in fact explicit that this is their conclusion, stating “that insular odour representations are more temporally stable” (note that the lack of such an explicit statement with regard to Figures 2C-E doesn’t change the fact that the authors are attempting to make an analogous point there), despite not having done a comparison between taste and odor coding. Once again, they need to do this analysis to reach this conclusion.

These are not small matters, because the authors’ interpretations of these panels are major parts of the contribution that the manuscript is meant to offer, and they have not YET been properly justified.

2) My only other central concern has to do with Figs S2B and S3. Fig S2B could use more description, and should be moved into the main document—the ruling out of alternative hypotheses is not a side issue to be relegated to Supplemental Material. I may be confused (because of the foreshortened description that goes with material so relegated), but I believe that

the appropriate test to rule out the confound that coding accuracy reflects hedonics rather than identity is simpler than what is shown in this panel, and easily within the authors' grasp: they should take each subject's pleasantness ratings, and show that their ability to predict odor from taste response (and vice-versa) does not depend on the two having similar hedonics that are distinct from the other odor and taste. Perhaps that is what the authors are doing with Fig S2B, but if so they haven't explained it clearly enough, and so I don't understand why the x-axis is "change in pleasantness between days." It would help to have more description ... which is, again, reason to move this panel to the main document, and give it more than one 3-line description in the text.

Relatedly, Fig S3 is currently referenced only in the Discussion section. This is inappropriate—the Discussion is meant for consideration of data that has already been extensively explained, either in the current or previous publications. If this analysis is worth discussing, it should be thoroughly unpacked in the Results section. If not, it should be eliminated.

Less central suggestions

It's not clear that "temporal instability" (Abstract) really describes the findings in rodent GC. That term makes the reader think of stochasticity and random change ... is that what the authors mean to suggest? Is that what the literature suggests?

Early on in the Results section, the authors make the point that "variation in individual-level responses did not affect perceived pleasantness or intensity at the group level." It's not clear what this means. It needs to be better explained.

The authors should more clearly explain (with references) the reason to believe that the volatiles used consistently evoke the perception of sweet/savory. Also, and more vitally, they need to reference their assumption that the retronasal odorants are truly taste-free, as their interpretations rest on this assumption.

Similarly, the authors need to reference their confidence that they are able to validly identify the dividing line between granular and a/dysgranular insular cortices.

With regard to the confusion matrices (3D-F), i.e., region-specific patterns of what gets confused for what, it would be a lot easier to see what's going on with an ancillary analysis presented in graph form.

Finally, some of the entries in the reference section are incomplete.

(Remarks on code availability)

Reviewer #2

(Remarks to the Author)

Summary: This study investigates how tastes and retronasal odors integrate in the brain to form a unified flavor perception, focusing on the role of the human insula. The researchers used functional MRI and a flavor-binding paradigm, where participants experienced congruent sweet and savory taste-odor pairs, followed by unimodal stimuli (tasteless odorants dissolved in solution or odorless tastants) during scanning sessions. They used multivariate pattern analysis (MVPA) to classify the neural patterns for odorants and tastes, and identified that they could cross-decode tastes and odorants in the insula, particularly in its ventral anterior subregion, suggesting crossmodal integration in this region. They ran analyses to examine taste and odorant representations over time, suggesting that taste representations in the insula were unstable over time, while odor representations remained stable. These findings support the idea that the insula contains a shared neural code for flavor identity and that it serves as the primary site for flavor integration, rather than downstream regions such as the orbitofrontal cortex.

General Comments: This is a well-designed and well executed study addressing a long-standing question in chemosensory neuroscience. I think the overall finding, that of successful cross-decoding of tastes and odorants in ventral anterior insula, is clear. I'm less confident in the strengths of some of other analyses, such as the granularity ROI analyses. I think the authors could also add a lot more detail to specific sections of the methods and results, which I have indicated below. I have also added some suggested analyses below that I think will add some clarity and strengthen these results. Regardless, I think this is a very strong paper. See specific comments below.

Specific Comments

1) Can you go into more detail about the purpose of the laboratory (non-scanning) session? I mostly understand why you did that session and made the flavor solutions, but I'm not totally clear and I think the casual reader would miss this entirely. It would seem the goal would be to build a cross-modal association between the taste and the odorant. How would you expect that association to change over the sessions? Do you think the results you observed would have been the same even without the laboratory training session?

2) Encoding and overlap claims: Frequently in this manuscript you claim that tastes and odors share overlapping encoding patterns, yet you don't ever directly test for this. The analyses that were done show that you can 'decode' both from the same region and that you can 'cross-decode' one from the other in the same region, but this is not the same thing. They are suggestive of a common code, but they don't directly demonstrate it. Pattern classification (or decoding) is a much more indirect measure than univariate activation. Above-chance classification can occur because of the absence of a signal as

well as the presence of one.

- I would suggest modifying your claims to better reflect what you showed with your analyses. Another option is running something like a voxel-based encoding model which might allow you to directly show that the voxels overlap.
- Importantly, claims about encoding are interpretations of the results and should be left for the Intro and Discussion sections. You should remove these claims from the Results section (eg. p.7, line 185. Your analysis shows that you can reliably classify from that region but not that it reliably encodes something).

3) Can you comment on the laterality of your results from the cross-modal searchlight analyses?

Methods

4) ROI Analyses: The ROI analyses, appear to aggregate results across both hemispheres of the brain. The granular partition analyses also appear to aggregate across both agranular and dysgranular regions of the insula, which appears to be about 2/3 of the area of the insula. It's not clear to me why you did this. A central goal of this study would seem to be providing some degree of specificity as to where flavor integration occurs in the brain. Running your ROI analyses in this manner would seem to run counter to this goal.

- The brainnetome atlas that you used for these analyses has 12 distinct granular, dysgranular, and agranular segments. I would suggest running and reporting your ROI analyses for these segments, or a subsection of them, as not all of these segments are typically associated with chemosensation. Tables would help, even if they are in the supplement.

- In particular, the gustatory ROI from figure 2A/B seems to include areas of both dorsal mid-insula and ventral anterior insula. So it's not clear which of the areas is most implicated in encoding. It would be helpful if you could clarify this point.

5) This lack of specificity with the ROI analyses is especially apparent for the cross-session analyses in Figure 4. What region of the brain are plots A,B, and C from? What ROIs were used for the analyses in 4D/E?

6) Did the univariate and MVPA analyses, which formed the basis of Fig 2 & 3, incorporate data from both sessions? How was this handled? Were data averaged across session? The methods section is very light on detail about this specifically.

7) Searchlights: The manuscript would greatly benefit from additional unimodal searchlight analyses to show where you observe above chance decoding for taste and odorants, and where they overlap.

- On that note, I would suggest using a searchlight radius of 3 voxels rather than 4 voxels for all the searchlight analyses (or at the very least showing whether it makes a difference or not). A 4-voxel radius means your searchlight sphere volume was ~256 voxels (>2000mm³, assuming you resampled to 2mm isotropic voxels during preprocessing [please clarify that, btw]). Combined with the multiple gaussian blurring steps used during pre and post-processing, this would mean that each of these searchlight spheres is bringing in information from a wide swath of brain tissue, hindering your efforts at specific localization.

8) On scan days, did participants' identification accuracy differ across chemosensory modalities or across sessions? Or both?

Minor Comments

9) P.3, Line 95: "Specifically, we test if object identity of retronasal taste-associated odours elicit dissociable patterns of activation in the gustatory cortex and if this encoding overlaps with the associated tastes." This sentence is extremely hard to parse. I would suggest condensing and rephrasing.

10) Please indicate chance level somewhere on the graphs for Figure 3 to assist readers trying to interpret them.

11) The area you identified as occipital cortex in Figure 3C looks much more like IPL/IPS. Please double check and clarify this.

12) I understand the intention of including the subthreshold areas in Figures 2 and 3, but without some kind of clear highlighting of the supra-threshold areas, it is very difficult to distinguish the two. I would suggest either outlining the supra-threshold areas in some way, or removing the sub-threshold areas entirely.

(Remarks on code availability)

Reviewer #3

(Remarks to the Author)

The study presents a highly intriguing and innovative approach to the question of odor-taste convergence. The methodology is particularly novel, offering fresh insights into the investigation of how food odors might activate taste-related areas in the brain following previous exposure to flavors, without relying on explicit semantic labels such as 'savory' or 'sweet.' This approach effectively encourages the encoding of taste and odor as flavors while minimizing the activation of semantic memories. Another notable strength of the study is the use of an MVPA classifier to predict odor decoding in the primary gustatory area. Additionally, the authors have made an important contribution by demonstrating stable encoding of flavor identity in the insula, while also revealing that the encoding of flavor in the orbitofrontal cortex (OFC) shows variability across days. These findings are both novel and valuable, offering new perspectives in the field.

While the study is definitely interesting, I do have two main concerns that I believe could be addressed to further strengthen

the manuscript.

1) What is the rationale behind selecting unimodal stimuli for testing instead of multimodal stimuli? It would be valuable to provide a justification for this choice, particularly considering that multimodal stimuli could reveal superadditivity effects. The results indicate convergence of odor and taste processing within a region of the insula. However, do these findings explicitly demonstrate an integrative mechanism?

2) A. Does the methodological approach effectively demonstrate early integration of odor and taste within taste-processing regions? To establish this, it would be necessary to examine brain connectivity, validating the primary integration of odor and taste in the insula before activation of the OFC. Please provide further arguments to support this interpretation or consider revising the discussion to present a more cautious conclusion. Also, kindly revise L22P11 accordingly.

B. The authors cite studies from Maiers' group to support the claim of early integration in the insula. However, Maiers' research indicates a relatively late neuronal response (approximately 1 second) in gustatory areas following odor presentation, compared to responses elicited by actual taste stimuli. Maier et al. suggested that their findings demonstrate convergence from other regions into the primary gustatory cortex (GC). While this could result from a direct connection between the GC and the olfactory cortex (OC), it may also be modulated by input from the OFC, or other top-down pathways. Clarification on this point would strengthen the argument.

Minor comments:

- L46P3: Please clarify the rationale for selecting this specific parcellation of the insula in the introduction. Providing justification for this choice would strengthen the methodological framework.
- L8P4: Please elaborate on the identification task, offering more details to ensure clarity and reproducibility.
- Results Section: Kindly include a reference to the complete table of results in the supplementary material for transparency and ease of access.
- MVPA Results: Is an accuracy of 54% indicative of a strong effect? It would be helpful to compare this result with similar MVPA findings in the literature to contextualize its significance.
- L21P8: The sentence "Conversely, a confusion matrix extracted from a spherical ROI around the IOFC peak shows that stimuli of the same flavour quality were equally likely to be confused for each other." requires clarification. Do you mean that both odor and taste stimuli associated with savory or sweet flavors were predicted by the same voxel pattern? The term confused refers to the confusion matrix but does not imply that the modality identities (odor vs. taste) were confused by the consumer—at least not solely due to activation in the OFC.
- L30P11: Please revise the following sentence for clarity: "This phenomenon has not previously been observed in other species, as previous experiments investigating the role of the insula in processing olfactory information have found no response a univariate activation or no crossmodal encoding between familiar tastes and odours." What do you mean by in other species? If you are referring to mammals other than humans, please ensure consistency with the cited references, as the current phrasing is ambiguous.
- L34P12: In the sentence "we propose that the insular patterns observed encoded identity, whereas the OFC patterns encoded value," please clarify: Do you mean that insular patterns encode flavor identity, while the OFC encodes pleasantness? Explicitly defining value in this context would improve clarity.
- L47P12: The term functional could be more precise; consider using electrophysiological to enhance clarity.
- L47P14: Please specify the exact question posed for each task (identity, pleasantness, and intensity) to ensure methodological transparency.
- L17-18P15: Could you confirm whether the accuracy score was calculated based on successful associations between flavors and their corresponding visual cues? Clarifying this point would aid interpretation.
- L33P16: Welcome appears to contain a typographical error (Wel(l)come). Please correct as needed.

(Remarks on code availability)

Version 1:

Reviewer comments:

Reviewer #1

(Remarks to the Author)

I am satisfied with the changes made by the authors.

(Remarks on code availability)

Reviewer #2

(Remarks to the Author)

The authors have responded to my comments and suggestions with sufficient clarity and detail. I believe the manuscript is greatly improved as a result and have no further edits, comments, or suggestions.

(Remarks on code availability)

Reviewer #3

(Remarks to the Author)

Thank you to the authors for addressing most of my previous concerns. I have a few follow-up comments to help finalize the reviewing process.

Follow-up on previous Comment 2.B:

Thank you for your explanation. The swallow breath indeed allows for a maximal number of odorants to reach the nasal cavity; however, depending on the quantity of liquid in the mouth (the less, the better), odorants can reach the olfactory epithelium even without swallowing.

Furthermore, regarding Maier et al.'s studies, since odor responses precede taste responses in the olfactory cortex (OC), and taste responses precede odor responses in the gustatory cortex (GC), the authors concluded that there are converging gustatory and olfactory inputs. However, they do not claim evidence of a direct anatomical connection.

In contrast, your brain connectivity data indicate connections between the piriform cortex and the insula, in the absence of top-down influence from the orbitofrontal cortex (OFC). This constitutes a compelling argument. Nevertheless, are the connections demonstrated by your connectivity analyses direct anatomical connections, or could they reflect indirect pathways? Please clarify this point.

Page 14, Discussion:

Concerning the following sentence:

"This phenomenon has not previously been observed in humans or other mammals, as previous experiments investigating the role of the insula in processing olfactory information have found no response^{8,35}, a univariate activation^{20,21,36} or no crossmodal encoding between familiar tastes and odours^{14,32}."

This statement is accurate regarding fMRI studies; however, Maier et al.'s work in rodents did demonstrate involvement of the gustatory cortex in odor processing. Please consider clarifying this distinction in the text to avoid any potential confusion.

Terminology clarification:

The expression "consummatory subjective value" remains ambiguous. It is unclear whether it refers specifically to liking, wanting, palatability, or a combination of these dimensions. Please provide a clearer definition or justification of the term's intended meaning within the context of your study.

Minor comments:

- Supplementary Figure S7:
 - o Please add a brief explanation of the terms "commonalities" and "crossmodal" in the figure legend for clarity.
 - o Please also specify in the legend that "PEB" refers to parametric empirical Bayes.

(Remarks on code availability)

Reviewer #1 (Remarks to the Author):

The work in this manuscript addresses a relatively simple question: if certain smells are described as “sweet” (or “savory”), will retronasal delivery of those smells activate gustatory cortex, and will that activation resemble the activation driven by (congruent) sweet taste delivery? The authors’ analyses of fMRI BOLD signals provide evidence supporting this hypothesis. They go on to provide evidence that: 1) this phenomenon is restricted to ventral gustatory insular cortex, and not to dorsal insular or olfactory piriform cortex; and that 2) the taste-smell similarity fades in comparisons made between sessions, which they interpret to mean that taste responses “drift” (more than olfactory responses).

There’s a lot to like about this brief report, and I am enthusiastic about seeing them added to the literature. The conclusions are nontrivial (general-language descriptors like “sweet” don’t necessarily refer to an actual taste; when we call a person “sweet,” we are seldom imagining a taste sensation), and for the most part (see minor quibbles below) the analyses seem rigorous and well done. I am particularly pleased with the way that the authors pre-specified specific hypotheses, and the careful, elegant use of the searchlight. There are only a few issues that I’d like to see addressed before I sign off on publication.

Authors’ response: We thank the reviewer for their positive evaluation of our study methodology and analyses. We share the reviewer’s enthusiasm for the results and their implications.

Central suggestions.

1) The authors repeatedly make an error in interpretation of their statistics that has been the topic of much discussion (see, for example, Nieuwenhuis, S. et al., Nat Neurosci, 14), wherein they perform separate analyses of two datasets, but then interpret their data as revealing significant differences BETWEEN the datasets. This is illegitimate: a significant effect found in only one of two datasets does not mean that the two datasets differ from one another; for instance, a $p = 0.04$ on one dataset and a $p = 0.06$ on the other will likely translate into a $p \sim 0.9$ with regard to the difference in the two effects.

The first of these errors has to do with Figures 2C-E. For the authors to convincingly reach the conclusion that they want to reach with these panels—which is that ventral insular codes odors and x-modality pairs BETTER than dorsal insular—they must do a direct comparison of the results for the 2 regions and show a difference. A multivariate ANOVA showing a significant region x stim-type interaction, followed by post-hoc comparisons, would do the job. Even better would be planned comparisons, which are (less conservative and) justified by References 23 and 27.

Similarly, the authors wish to convince the reader that odor coding is more stable across days than taste coding (Figures 4A-C). In this case they are in fact explicit that this is their conclusion, stating “that insular odour representations are more temporally stable” (note that the lack of such an explicit statement with regard to Figures 2C-E doesn’t change the fact that the authors are attempting to make an analogous point there), despite not having done a comparison between taste and odor coding. Once again, they need to do this analysis to reach this conclusion.

These are not small matters, because the authors' interpretations of these panels are major parts of the contribution that the manuscript is meant to offer, and they have not YET been properly justified.

Authors' response: We thank the reviewer for pointing out the lack of statistical justification for the conclusions we draw about differences between ROIs. We have now performed pairwise comparisons of all the decoding accuracies between the two ROIs, in addition to modifying **Figure 2** to better reflect this:

From Page 7 (Results):

Critically, accuracy for the crossmodal decoder was significantly higher for the dysgranular and agranular ROI than for the granular insula ROI ($t(24) = 3.208, p = .006$; **Figure 2D**). For taste ($t(24) = 0.292, p = .760$) and odour decodability ($t(24) = 1.288, p = .195$) no such difference between the ROIs was observed.

For the across-day analyses, we agree that our analyses do not justify the conclusion of differences between the experimental conditions. We have now revised the statements mentioned by the reviewer to reflect that we found significant interaction effects in taste decoding, without implying that the effects in odour or crossmodal decoding are significantly lower. We have also opted to use linear mixed-effects models (LMMs) over our previous two-way ANOVAs to take into account the fact that the data we are comparing are not independently and individually distributed (iid):

From Page 12 (Results):

Confirming previous observations of temporal change in taste identity representations, decoder performance for taste quality dropped significantly when the decoder was trained and tested on different days (**Figure 5A**; LMM $\beta = 0.12, p = .044$), indicating that pattern-based encoding of taste qualities in the insula is not stable across days. This temporal sensitivity effect was not significant for the decoder trained and tested on odour quality or crossmodal decoding (**Figure 5B – C**).

2) My only other central concern has to do with Figs S2B and S3. Fig S2B could use more description, and should be moved into the main document—the ruling out of alternative hypotheses is not a side issue to be relegated to Supplemental Material. I may be confused (because of the foreshortened description that goes with material so relegated), but I believe that the appropriate test to rule out the confound that coding accuracy reflects hedonics rather than identity is simpler than what is shown in this panel, and easily within the authors' grasp: they should take each subject's pleasantness ratings, and show that their ability to predict odor from taste response (and vice-versa) does not depend on the two having similar hedonics that are distinct from the other odor and taste. Perhaps that is what the authors are doing with Fig S2B, but if so they haven't explained it clearly enough, and so I don't understand why the x-axis is "change in pleasantness between days." It would help to have more description ... which is, again, reason to move this panel to the main document, and give it more than one 3-line description in the text.

Authors' response: We thank the reviewer for this comment and agree that it is a crucial argument to make. We have therefore performed the control analyses mentioned by taking the absolute difference between the means of the pleasantness ratings of the sweet stimuli and those of the savoury stimuli. We have then tried to correlate this against crossmodal accuracy, which can be seen in both the revised **Figure 2** and the Results section. We have highlighted that we ruled out this alternative hypothesis in the Discussion section as well.

From Page 7 (Results):

To rule out the possibility that these decoding accuracies were driven by hedonic differences, we compared the resultant crossmodal accuracy against absolute differences in hedonic ratings of each participant between sweet and savoury stimuli. We found no correlation between the hedonic differences and crossmodal decoding performance (**Figure 2E**).

From Page 14 (Discussion):

Importantly, our results cannot be explained by a mere overlap in hedonic features, as between-participant variability of hedonic differences did not predict decoder performance. They are also robust to changes in the exact identity of the odours chosen, which varied across participants.

Relatedly, Fig S3 is currently referenced only in the Discussion section. This is inappropriate—the Discussion is meant for consideration of data that has already been extensively explained, either in the current or previous publications. If this analysis is worth discussing, it should be thoroughly unpacked in the Results section. If not, it should be eliminated.

Authors' response: We thank the reviewer for this comment. We have now mentioned it in the Results section prior to its discussion in the Discussion section (note that it is now referred to as **Supplementary Figure S6**):

From Page 10 (Results):

Notably, we did not observe above-chance decoder performance for the same partitions in the piriform cortex in either the whole-brain searchlight analysis or the restricted ROI analysis (**Supplementary Figure S6**)

Less central suggestions

It's not clear that "temporal instability" (Abstract) really describes the findings in rodent GC. That term makes the reader think of stochasticity and random change ... is that what the authors mean to suggest? Is that what the literature suggests?

Authors' response: Thank you for this comment—we agree with the reviewer that we did not mean to refer to a stochastic/random change but rather to a degradation over time, which may very well be systematic in nature. We have changed the term to 'temporal drift' to clarify this.

From Page 2 (Abstract):

Additionally, we observed temporal drift in insular taste representations, paralleling findings in rodent gustatory cortex.

Early on in the Results section, the authors make the point that "variation in individual-level responses did not affect perceived pleasantness or intensity at the group level." It's not clear what this means. It needs to be better explained.

Authors' response: We apologise for not being clear. What we meant to state here was that it can be seen from the supplementary figure that the intra- and interpersonal variability was comparable across stimulus categories and did not result in systematic differences between them.

We have now clarified the sentence to state:

From Page 4 (Results):

As seen in **Supplementary Figures S2B and S2C**, we observed intra- and interpersonal variability in perceived intensity and pleasantness ratings within each of the stimulus categories. However, these differences did not vary systematically between stimulus categories and thus did not cause any systematic differences between them at the group level.

The authors should more clearly explain (with references) the reason to believe that the volatiles used consistently evoke the perception of sweet/savory. Also, and more vitally, they need to reference their assumption that the retronasal odorants are truly taste-free, as their interpretations rest on this assumption.

Authors' response: We thank the reviewer for this comment and agree with the importance of this point. For that reason, we piloted the dissolved odorants with nose-clipped volunteers who attested to their tastelessness. We have now also conducted stringent triangle testing of the dissolved odorants against artificial saliva, which we have detailed in the Methods and Materials section, as well as **Supplementary Figure S1**.

From Page 19 (Methods):

During piloting, pure odorants dissolved in ArtS were tested by nose-clipped volunteers and asked for verbal descriptions of taste sensation, whereupon volunteers attested to their tastelessness. Additional triangle testing of the odorant solution was performed to ensure tastelessness. Three taste cups containing ArtS were presented to nose-clipped volunteers, where two contained only ArtS and one the odorant dissolved in ArtS. This process was repeated for eight trials, such that a binomial test could be carried out, where six of eight correct identifications would yield a significant one-tailed binomial test at $\alpha < .05$. This technique has previously been used to demonstrate lack of discriminability⁶⁹. In total, five volunteers performed the task, where four volunteers tested four odorants, and one volunteer tested two odorants. Therefore, each odorant was tested by three volunteers.

Supplementary Figure S1 shows that the nose-clipped volunteers performed at around chance level, and no volunteer could significantly distinguish the odorant from plain ArtS solution.

In addition, our design ensured that the stimuli consistently evoked their respective flavour association. During the laboratory training session, participants had to learn abstract visual cues that corresponded to the flavours (sweet and savoury) through an identification task with feedback. Prior to each scan session, participants again performed the identification task, in addition to pleasantness and intensity ratings. Participants could reliably distinguish the stimuli and classify them as 'sweet' and 'savoury' using the visual cues they were trained on. We refrained from using those words to describe the flavours to avoid potential semantic biases and priming effects. Accuracy scores can be seen in **Supplementary Figure S3**.

From Page 4 (Results):

Supplementary Figure S3 shows the session-wise identification accuracy per stimulus. A linear mixed-effects model (LMM) of the accuracy showed neither main effects of session or flavour nor interaction effects of the same (**Supplementary Table S1**).

From Page 18 (Methods):

Prior to the session, participants were randomly assigned two abstract visual cues (letters from the Phoenician alphabet), one for the savoury flavour and another for the sweet flavour. In each trial of the rating task, the participant was presented 1 ml of a stimulus and was asked to choose which visual cue it corresponded to, such that participants would learn the cue associated with each flavour. **During this task, the word 'Identify' was presented at the top of the screen, and participants could choose between the two visual cues or decline to answer, which would lead to no response.** This cue association allowed us to examine if participants were able to distinguish the flavours without being primed by words such as 'sweet' or 'savoury'.

Similarly, the authors need to reference their confidence that they are able to validly identify the dividing line between granular and a/dysgranular insular cortices.

Authors' response: The Brainnetome atlas by Fan et al. (2016)¹ uses a combination of probabilistic diffusion tractography, connectivity-based parcellation, cross-validated optimal clustering, anatomical connectivity patterns and resting-state functional connectivity patterns from a subset of the Human Connectome Project data. This results in a probabilistic map of the standard Montreal Neurological Institute template of various myelo-cytoarchitectural structures. The authors have demonstrated that their atlas subdivisions map well onto cortical activity associated with the respective anatomical area, indicating that, while probabilistic in nature, it captures meaningful neuroanatomical differences appropriately. As our data were converted to the standard MNI template, we were able to use these probabilistic parcellations to delineate the granular and a/dysgranular insular cortices. For the present study, we chose to combine several sub-regions of the atlas to form our ROIs. Specifically, our granular insular cortex ROI was comprised of the bilateral hypergranular insula

(areas 163 and 164) and the bilateral dorsal granular insula (areas 171 and 172), whereas the a/dysgranular cortices consisted of the remainder of the insula. We reasoned that conducting our analyses on these aggregated ROIs would limit idiosyncratic differences in myelo-cytoarchitecture, ensuring that our parcellation between the granular and the a/dysgranular cortices would indeed be reflective of their respective granularity. We have added this justification to the paper as well. See also our answer to Reviewer #2, question 4.

From Page 21 (Materials and Methods)

Post-hoc ROI MVPA used subregions of the insula as parcellated in the Brainnetome Atlas by Fan et al. (2016), which uses a combination of probabilistic diffusion tractography, connectivity-based parcellation, cross-validated optimal clustering, anatomical connectivity patterns and resting-state functional connectivity patterns from a subset of the Human Connectome Project data. To limit idiosyncratic differences in myelo-cytoarchitecture, atlas subregions 163/164 and 171/172, were merged into a granular ROI, whereas the rest of the insula formed the dysgranular/agranular ROI.

With regard to the confusion matrices (3D-F), i.e., region-specific patterns of what gets confused for what, It would be a lot easier to see what's going on with an ancillary analysis presented in graph form.

Authors' response: We have now added ancillary analyses of the data in the confusion matrices in **Supplementary Figure S8**.

Finally, some of the entries in the reference section are incomplete.

Authors' response: We have carefully reviewed the reference list and exchanged reference Nr. 19 for a full citation to the published paper instead of OSF, in addition to adding the missing journal name for reference Nr. 23.

Reviewer #2 (Remarks to the Author):

Summary: This study investigates how tastes and retronasal odors integrate in the brain to form a unified flavor perception, focusing on the role of the human insula. The researchers used functional MRI and a flavor-binding paradigm, where participants experienced congruent sweet and savory taste-odor pairs, followed by unimodal stimuli (tasteless odorants dissolved in solution or odorless tastants) during scanning sessions. They used multivariate pattern analysis (MVPA) to classify the neural patterns for odorants and tastes, and identified that they could cross-decode tastes and odorants in the insula, particularly in its ventral anterior subregion, suggesting crossmodal integration in this region. They ran analyses to examine taste and odorant representations over time, suggesting that taste representations in the insula were unstable over time, while odor representations remained stable. These findings support the idea that the insula contains a shared neural code for flavor identity and that it serves as the primary site for flavor integration, rather than downstream regions such as the orbitofrontal cortex.

General Comments: This is a well-designed and well executed study addressing a long-standing question in chemosensory neuroscience. I think the overall finding, that of successful cross-decoding of tastes and odorants in ventral anterior insula, is clear. I'm less confident in the strengths of some of

other analyses, such as the granularity ROI analyses. I think the authors could also add a lot more detail to specific sections of the methods and results, which I have indicated below. I have also added some suggested analyses below that I think will add some clarity and strengthen these results. Regardless, I think this is a very strong paper. See specific comments below.

Authors' response: We appreciate the reviewer's kind comments of our study design and execution. We are happy to hear that they consider this a strong manuscript, and we are thankful for their comments that have improved it.

Specific Comments

1) Can you go into more detail about the purpose of the laboratory (non-scanning) session? I mostly understand why you did that session and made the flavor solutions, but I'm not totally clear and I think the casual reader would miss this entirely. It would seem the goal would be to build a cross-modal association between the taste and the odorant. How would you expect that association to change over the sessions? Do you think the results you observed would have been the same even without the laboratory training session?

Authors' response: The laboratory session served multiple purposes, of which inducing flavour binding was one. Due to potential cultural differences in how odours are perceived (e.g. saffron is typically associated with savoury foods in Mediterranean and Middle Eastern countries but with sweet foods in Sweden), we could not take potential individual variations arising from prior exposures for granted. Therefore, we deemed a laboratory training session appropriate and necessary. Participants' identification accuracies (of whether the stimulus they received were perceived to be sweet or savoury, which they chose using the abstract visual cues they were randomly assigned to) can be seen in **Supplementary Figure S3**, and these accuracies did not change across sessions, as we have now elaborated in the Results section:

From Page 4 (Results):

Supplementary Figure S3 shows the session-wise identification accuracy per stimulus. A linear mixed-effects model (LMM) of the accuracy showed neither main effects of session or flavour nor interaction effects of the same (**Supplementary Table S1**).

For clarity, we have now elaborated this in the Methods section:

From Pages 17 – 18 (Methods):

Laboratory session. Participants were trained on the task and the stimuli pairing simultaneously in their pre-scanning behavioural session. **This session served multiple purposes:**

- It allowed the selection of the specific odorants used for each participant, ensuring approximate isointensity.
- It ensured that our results were generalisable, as we selected different odorant combinations (see **Supplementary Figure S2A**).

- We trained participants on taste-odour pairings by presenting them taste-odour combinations in repetition, as taste-odour associations can be culture-specific, and we recruited from an international population of English speakers in Stockholm.
- Participants learned the abstract symbols representing the sweet and savoury flavours, ensuring that we could test stimulus identification without relying on potential biases arising from explicit semantic labels such as 'sweet' or 'savoury'.
- Participants were trained to consume small aliquots of liquid (1 ml), delivered via a similar setup as in the scanner, while in a supine position on a massage bench, which minimised discomfort and the potential for choking during the actual scan.

2) *Encoding and overlap claims: Frequently in this manuscript you claim that tastes and odors share overlapping encoding patterns, yet you don't ever directly test for this. The analyses that were done show that you can 'decode' both from the same region and that you can 'cross-decode' one from the other in the same region, but this is not the same thing. They are suggestive of a common code, but they don't directly demonstrate it. Pattern classification (or decoding) is a much more indirect measure than univariate activation. Above-chance classification can occur because of the absence of a signal as well as the presence of one.*

- I would suggest modifying your claims to better reflect what you showed with your analyses. Another option is running something like a voxel-based encoding model which might allow you to directly show that the voxels overlap.

Authors' response: We thank the reviewer for pointing out this issue. We have conducted additional supplementary analyses to show that crossmodal decoding accuracy could be predicted by overlapping tuning indices between the taste and odour modalities, which indicates that our decoding results are suggestive of similarities in the taste and odour tuning maps. These results are described in the Results section as well as **Figure 3**:

From Pages 7-8 (Results):

In order to ensure that the above-chance decoding we observed in our ROIs are indeed reflective of overlapping activation patterns, we extracted the tuning index for each voxel to each modality by contrasting their statistical parametric maps (see **Methods** and **Figure 3A**). We found that tuning index overlap, derived from crossmodal correlations of tuning index maps, were predictive of the performance of the crossmodal decoder in the leave-one-subject-out gustatory ROI ($r = .507, p = .010$; **Figure 3B**), as well as the anatomically defined parcellations of the granular insula ($r = .557, p = .004$; **Figure 3C**) and dysgranular and agranular insula ($r = .535, p = .006$; **Figure 3D**). This indicates that above-chance crossmodal decoding was indeed driven by overlapping flavour-specific patterns of activations between the modalities.

- Importantly, claims about encoding are interpretations of the results and should be left for the Intro and Discussion sections. You should remove these claims from the Results section (eg. p.7, line 185. Your analysis shows that you can reliably classify from that region but not that it reliably encodes something).

Authors' response: We thank the reviewer for this comment and have now moved all references to encoding to the Discussion section. The Results section now states the decodability of flavour identity, while the Discussion section explicitly mentions that we interpret the decodability of tastant identity as spatial encoding of taste identity.

3) *Can you comment on the laterality of your results from the cross-modal searchlight analyses?*

Authors' response: We thank the reviewer for raising this interesting question. Our results indeed seem to indicate preferential processing of crossmodal overlap in the right cerebral hemisphere, but we would like to approach this question with caution for a number of reasons:

- While our supra-threshold findings seem to indicate a lateralisation to the right hemisphere, it is important to consider this cutoff in the context of the underlying pattern of more widely distributed data. Subthreshold patterns (indicated by the translucent overlays throughout the manuscript) indicate that decodability patterns between the right and left hemisphere mirror each other to a large extent but fail to survive stringent correction for multiple comparisons in the left hemisphere.
- Hemispheric lateralisation is known to be characterised by large interpersonal variability and to be related to handedness²; for the current study, we neither selected our participants based on indices of such lateralisation, nor did we collect data that would be indicative of lateralisation strength such as a handedness inventory which could be related to the findings in a post-hoc analysis. It is therefore impossible to speculate to what extent the lateralisation observed would generalise to the general population.
- Investigation of hemispheric lateralisation of perceptual processing has been the topic of intense scientific investigation for decades, but conflicting results continue to be reported³⁻⁵ and, as such, the relevance of such potential differences remains currently unresolved. For this reason, even if we were confident that the observed lateralisation was systematic and meaningful, we do not currently have a strong theoretical foundation to build on to develop a meaningful interpretation of such an effect.

We have now addressed this question in the discussion:

From Page 15 (Discussion):

While inspection of subthreshold data largely indicates symmetrical activation patterns, suprathreshold decoding peaks were predominantly located in the right hemisphere. Given the lack of data on cortical lateralisation in the current participant sample, whether this represents a systematic processing difference remains to be clarified.

Methods

4) *ROI Analyses: The ROI analyses, appear to aggregate results across both hemispheres of the brain. The granular partition analyses also appear to aggregate across both agranular*

and dysgranular regions of the insula, which appears to be about 2/3 of the area of the insula. It's not clear to me why you did this. A central goal of this study would seem to be providing some degree of specificity as to where flavor integration occurs in the brain. Running your ROI analyses in this manner would seem to run counter to this goal.

Authors' response: As the Brainnetome atlas we used relied on probabilistic estimates of specific myelo-cytoarchitectural properties of their parcellations, we combined several sub-regions to form our ROIs. Specifically, the granular insular cortex was comprised of the bilateral hypergranular insula (areas 163 and 164) and the bilateral dorsal granular insula (areas 171 and 172), whereas the a/dysgranular cortices consisted of the remainder of the insula. The probabilistic nature of the atlas implied a level of uncertainty of individual granularity. Therefore, we reasoned that conducting our analyses on these aggregated ROIs would limit idiosyncratic differences in myelo-cytoarchitecture, ensuring that our parcellation between the granular and the a/dysgranular cortices would indeed be reflective of their respective granularity. See also our answer to Reviewer #1.

- The brainnetome atlas that you used for these analyses has 12 distinct granular, dysgranular, and agranular segments. I would suggest running and reporting your ROI analyses for these segments, or a subsection of them, as not all of these segments are typically associated with chemosensation. Tables would help, even if they are in the supplement.

Authors' response: We thank the reviewer for this suggestion. We have now run MVPA decoding on all parcellations provided in the Brainnetome atlas, and the results can be seen in **Supplementary Figure S5** (see p.7 of the main manuscript). These results largely agree with what we report for the larger subdivisions.

- In particular, the gustatory ROI from figure 2A/B seems to include areas of both dorsal mid-insula and ventral anterior insula. So it's not clear which of the areas is most implicated in encoding. It would be helpful if you could clarify this point.

Authors' response: We apologise for being unclear about the analysis strategy underlying the presentation of our results. Our original preregistered analysis plan was based on a functional definition of canonical gustatory cortex, which yielded the ROI seen in **Figure 2A–B** that the reviewer is referring to, and which was derived from a pre-registered leave-one-subject-out cross-validation of the univariate taste-ArtS contrast. This stringent approach ensured no information leakage between the ROI generation and the data used for MVPA, although this resulted in ROIs covering both dorsal mid-insula and ventral anterior insula, in addition to high variability in insular coverage (**Supplementary Figure S4**).

As the reviewer points out, the resulting ROI was rather large and spanned different parcellations of insular cortex. We therefore suspected that the observed activation may result from functionally separate subclusters of activation, which would further be in line with the literature on piriform projections to the dysgranular and agranular insula.

We therefore decided to expand our original pre-registered analysis plan with an exploratory analysis, which further parcellate the insula based on probabilistic mapping of the cytoarchitecture using the Brainnetome atlas and allowed us to conduct further analyses using these anatomically defined ROIs. We interpret our results in line with the hypothesis stated above: while we were able to successfully decode odour and taste information from the combined functionally defined cluster, the exploratory follow-up analyses indicate that we might have actually captured activation from separable areas where the anterior part contains crossmodal representations in line with known anatomical projections, while the more posterior part may be more restricted to taste encoding.

We have now clarified this procedure and our reasoning behind it:

From Page 7 (Results):

[...] we performed a pre-registered multivariate pattern analysis (MVPA) decoding in independent functional ROIs in the bilateral insula responsive to tastants, generated using leave-one-subject-out cross-validation (see **Methods** and **Supplementary Figure S4**).

The functional ROI generated from the leave-one-subject-out method included both dorsal mid-insula and ventral anterior insula, which extend beyond areas associated with primary gustatory cortex. These range from granular to agranular regions and are slightly different for each subject (**Supplementary Figure S4**). Given that insular projections from the piriform cortex largely terminate in its dysgranular and agranular sections in both primates and rodents^{23,27}, we suspected that our functional ROI captured functionally separate parcellations of the insular cortex. In an exploratory follow-up analysis, we tested for differential identity-specific responsiveness to flavour components in sub-regions parcellating the ROI based on granularity (**Figure 2C**) using a probabilistic anatomical map²⁸ to show the specific sub-regions showing overlapping patterns for tastes and odours.

5) *This lack of specificity with the ROI analyses is especially apparent for the cross-session analyses in Figure 4. What region of the brain are plots A, B, and C from? What ROIs were used for the analyses in 4D/E?*

Authors' response: We apologise for the lack of clarity in the manuscript about this, which we realise may create difficulties in understanding for the reader. We have now made sure to clarify where each ROI came from in our analyses. What was previously **Figure 4** is now **Figure 5**, and we have made clear that plots A, B and C came from our leave-one-subject-out functional ROI. This analysis was one of our pre-registered analyses, with this leave-one-subject-out functional ROI as our pre-registered ROI. Subsequent analyses that incorporate RSA were exploratory and used our searchlight peaks, as those coordinates were likely to have the most robust signal.

From Page 12 (Results):

Specifically, we compared data trained and tested from the same scanning session against those trained and tested from different sessions in the leave-one-subject-out ROI (**Supplementary Figure S4**).

Figure 5D shows baseline representational drift (as judged by the representational drift of the same flavour across runs) in both the insula and the OFC ROIs, defined as a 10 mm radius sphere from the searchlight peaks in these respective regions.

6) Did the univariate and MVPA analyses, which formed the basis of Fig 2 & 3, incorporate data from both sessions? How was this handled? Were data averaged across session? The methods section is very light on detail about this specifically.

Authors' response: Univariate and MVPA analyses incorporated data from all runs (barring runs that had more than 10% scrubbed frames due to excessive movement). We used the default SPM design, where each run had a runwise intercept, in addition to applying grand mean scaling. We have added the following sentence to both parts of the Methods section to clarify this:

From Pages 21 and 22 (Methods):

The design matrix included a runwise intercept and applied the default SPM grand mean scaling, such that the mean over space and time within a run were scaled to 100.

7) Searchlights: The manuscript would greatly benefit from additional unimodal searchlight analyses to show where you observe above chance decoding for taste and odorants, and where they overlap.

- On that note, I would suggest using a searchlight radius of 3 voxels rather than 4 voxels for all the searchlight analyses (or at the very least showing whether it makes a difference or not). A 4-voxel radius means your searchlight sphere volume was ~ 256 voxels ($>2000\text{mm}^3$, assuming you resampled to 2mm isotropic voxels during preprocessing [please clarify that, btw]). Combined with the multiple gaussian blurring steps used during pre and post-processing, this would mean that each of these searchlight spheres is bringing in information from a wide swath of brain tissue, hindering your efforts at specific localization.

Authors' response: We thank the reviewer for these suggestions. We have conducted unimodal searchlight analyses, which can be found in **Supplementary Table S5**:

From Page 10 (Results):

Supplementary searchlight results of tastant identity decoding can be found in **Supplementary Table S5**, whereas searchlight results of odour identity decoding did not survive TFCE.

Furthermore, we have re-done all searchlight analyses using a radius of 3 voxels, as opposed to the 4 voxels that were closer to our pre-registered searchlight radius of 9 mm. All searchlight results presented now come from a spherical searchlight of 3 voxels.

In addition, we have clarified that all functional images were resampled to a 2 mm isotropic resolution.

8) On scan days, did participants' identification accuracy differ across chemosensory modalities or across sessions? Or both?

Authors' response: No such differences were observed, and these data are now presented in **Supplementary Figure S3**. Please see response to Comment 1 for additional detail.

Minor Comments

9) P.3, Line 95: "Specifically, we test if object identity of retronasal taste-associated odours elicit dissociable patterns of activation in the gustatory cortex and if this encoding overlaps with the associated tastes." This sentence is extremely hard to parse. I would suggest condensing and rephrasing.

Authors' response: We thank the reviewer for this comment. The sentence now reads as follows:

From Page 3 (Results)

Specifically, we test if retronasal taste-associated odours elicit dissociable patterns of activation in the gustatory cortex and if **these patterns** overlap with **those of** the associated tastes.

10) Please indicate chance level somewhere on the graphs for Figure 3 to assist readers trying to interpret them.

Authors' response: Thank you for this comment. We have now highlighted the above-chance predictions on the confusion matrices on **Figure 4** (previously **Figure 3**) in bold. We have also stated that chance level is at 25% in the figure caption. Furthermore, chance-level lines can be found in the ancillary analyses of the confusion matrices on **Supplementary Figure S8**.

11) The area you identified as occipital cortex in Figure 3C looks much more like IPL/IPS. Please double check and clarify this.

Authors' response: We thank the reviewer for pointing this out. We have double-checked it and agree that it is indeed the IPL. We have now changed this in the manuscript:

From Page 10 (Results):

However, the highest crossmodal decoder performance was observed in a region extending from the **inferior parietal lobule (IPL)** to the bilateral cuneus (peak $z = 2.82$, $p_{TFCE} = .002$), an area that does not typically form part of the flavour network.

Furthermore, for consistency, we have now elected to use the labels provided by the Brainnetome Atlas to label the peaks of our crossmodal searchlight analyses, seen in **Supplementary Table S4**.

12) *I understand the intention of including the subthreshold areas in Figures 2 and 3, but without some kind of clear highlighting of the supra-threshold areas, it is very difficult to distinguish the two. I would suggest either outlining the supra-threshold areas in some way, or removing the sub-threshold areas entirely.*

Authors' response: We thank the reviewer for this comment. We believe that showing subthreshold areas is important for scientific transparency and openness. However, we have now outlined the suprathreshold regions of **Figures 2 and 4** in red, making the distinction between the regions more visible.

Reviewer #3 (Remarks to the Author):

The study presents a highly intriguing and innovative approach to the question of odor-taste convergence. The methodology is particularly novel, offering fresh insights into the investigation of how food odors might activate taste-related areas in the brain following previous exposure to flavors, without relying on explicit semantic labels such as 'savory' or 'sweet.' This approach effectively encourages the encoding of taste and odor as flavors while minimizing the activation of semantic memories. Another notable strength of the study is the use of an MVPA classifier to predict odor decoding in the primary gustatory area. Additionally, the authors have made an important contribution by demonstrating stable encoding of flavor identity in the insula, while also revealing that the encoding of flavor in the orbitofrontal cortex (OFC) shows variability across days. These findings are both novel and valuable, offering new perspectives in the field.

Authors' response: We value the reviewer's appreciation of our approach to this pertinent question. We are glad that the reviewer believes the study to be an important contribution.

While the study is definitely interesting, I do have two main concerns that I believe could be addressed to further strengthen the manuscript.

1) *What is the rationale behind selecting unimodal stimuli for testing instead of multimodal stimuli? It would be valuable to provide a justification for this choice, particularly considering that multimodal stimuli could reveal superadditivity effects. The results indicate convergence of odor and taste processing within a region of the insula. However, do these findings explicitly demonstrate an integrative mechanism?*

Authors' response: We agree that the comparison of multimodal stimuli to unimodal stimuli is essential for investigating superadditive mechanisms of flavour integration. However, we would like to stress that this question was beyond the scope of this study. Our goal here was to delineate the brain areas that are characterised by overlapping responses to unimodal percepts linked to the same flavour quality, specifically concerning taste sensations elicited by retronasal odours without concurrent taste stimulation. In this particular situation, presenting multimodal stimuli would introduce a confounding factor, as we would no longer be able to tell which modality would contribute to the observed signal. That said, upon re-reading our manuscript, we realised that we have not always clearly distinguished between crossmodal association and crossmodal

integration, and we understand that this can confuse the reader. We have carefully reviewed the manuscript for language concerning integration, and hope this will clarify our research approach for the reader.

From Page 2 (Abstract):

These findings underscore the robust **crossmodal influences** of gustatory and retronasal olfactory processing that underpin the flavour percept.

From Page 3 (Introduction):

Overlapping representations of smell and taste may occur prior to the interaction of the chemosensory percept with other sensory properties of foods in the OFC.

Here, we use oral delivery of unimodal chemosensory stimulation (tasteless odorants and odourless tastants; **Figure 1A**) and functional magnetic resonance imaging (fMRI) to **investigate early overlap in processing of chemosensory information** in the insula.

From Page 8 (Results):

These results highlight the insula's unique role in flavour perception, starting from processing of basic tastant quality in the granular insula and its integration **into a shared representation with odour quality that likely underlies the formation of a flavour percept** in the dysgranular and agranular insula.

From Page 14 (Discussion):

Our findings highlight the role of the insula as a potential hub **for the representation of conceptual overlap between flavour components**. This putative early integration explains the strong perceptual link between tastes and odours and can be explained by simultaneous stimulation arising from both modalities during food consumption.

From Page 15 (Discussion):

Nevertheless, **early overlapping chemosensory representations in the insula do not** necessarily rule out the role of the OFC in evaluating the subjective value of food reward.

We agree with the reviewer that, for future studies, it would certainly be interesting to specifically test whether the same region of ventral insula identified for cross-modal overlap in flavour representations, also are an important contributor to integration of unimodal signals during their binding into a multisensory flavour percept. This could, for example, explain the cross-modal sensory enhancement observed when smell and taste are concurrently presented. We believe that our study, which demonstrates that odours and tastes can evoke the same patterns even when presented in isolation from each other, provides an important conceptual foundation for such future research, and now highlight this as a potential avenue for future studies.

From P.17 (Discussion):

Our work also provides a theoretical foundation to further explore the mechanisms of odour-induced taste enhancement, which arise when a conceptually matching odorant and tastant

are presented concurrently. Our data indicate a potential functional mechanism by which this mutual enhancement could be implemented, but testing it on multisensory stimulus combinations would have introduced confounds and was beyond the scope of this study.

2) A. *Does the methodological approach effectively demonstrate early integration of odor and taste within taste-processing regions? To establish this, it would be necessary to examine brain connectivity, validating the primary integration of odor and taste in the insula before activation of the OFC. Please provide further arguments to support this interpretation or consider revising the discussion to present a more cautious conclusion. Also, kindly revise L22P11 accordingly.*

Authors' response: We thank the reviewer for this comment. While an fMRI approach does not have the temporal resolution required to show that decodability in the dys/agranular insula precedes that in the OFC, we have conducted additional Dynamic Causal Modelling (DCM) analyses on our ROIs using a Bayesian pruning of estimated parameters. In modelling the connection strengths, we were able to find that connectivity from the dys/agranular insula (dalns) to the lateral OFC was predictive of crossmodal accuracy in the dalns, but the converse connection was not predictive of it (**Supplementary Figure S7**). This implies that decodability in the dalns is not dependent on, and consequently not due to, information transfer from the IOFC. We have also added the following sentence to the Results section:

From Page 10 (Results):

Supplementary connectivity analyses indicated that crossmodal accuracy scores in the dysgranular/agranular insula were explained by its effective connectivity to the IOFC, but not by the reverse connection (see **Methods** and **Supplementary Figure S7**).

B. *The authors cite studies from Maier's group to support the claim of early integration in the insula. However, Maier's research indicates a relatively late neuronal response (approximately 1 second) in gustatory areas following odor presentation, compared to responses elicited by actual taste stimuli. Maier et al. suggested that their findings demonstrate convergence from other regions into the primary gustatory cortex (GC). While this could result from a direct connection between the GC and the olfactory cortex (OC), it may also be modulated by input from the OFC, or other top-down pathways. Clarification on this point would strengthen the argument.*

Authors' response: We thank the reviewer for bringing up this interesting point. It is true that, in the studies cited, GC neuronal responses to aqueous odours peaked at around 1 second after stimulus onset, which occurs much later than responses to actual taste stimulation which peaks at 0.5 seconds. Nevertheless, we argue that the different physiological mechanisms underlying taste and retronasal odour transduction and consequent perception may be behind this latency. Taste transduction likely occurs upon intraoral stimulus delivery, as that is when tastants make contact with taste receptors in the oral cavity. However,

retronasal odour transduction likely only occurs upon swallowing as opposed to stimulus delivery, as that is when odorants from the oral cavity can reach olfactory receptors. This delay is likely further exacerbated by the fact that rodents are obligate nose breathers⁶, which would prevent any odorants in the oral cavity from reaching the olfactory epithelium prior to swallowing. Such a delay in neuronal response is likely driven by a delay in transduction. For comparison, we point to the results shown in Figure 4 of Maier et al. (2017)⁷ indicating latency in GC responses to intraoral delivery of aqueous odours. Within the same figure, we also see a similarly delayed response in the olfactory cortex (OC) of rodents to the same stimuli. While a delayed response of up to 1 s would typically be indicative of modulation by other top-down pathways, our understanding of the physiological mechanisms behind retronasal odour transduction, as well as the fact that the primary olfactory cortex displays a comparable latency, strongly suggests a direct connection between the OC and the GC.

Minor comments:

• *L46P3: Please clarify the rationale for selecting this specific parcellation of the insula in the introduction. Providing justification for this choice would strengthen the methodological framework.*

Authors' response: We thank the reviewer for this comment. While we had highlighted in the introduction that afferent piriform projections to the insula were limited to the dysgranular and agranular portions, we have now tied it to our analyses more clearly.

From Page 3 (Introduction):

Crucially, after parcellating the insula based on layer morphology (granularity), we show that this crossmodal overlap is observed mostly in the ventral anterior subregion of the insula, which corresponds to areas of lower granularity, **aligning with abovementioned monosynaptic piriform-insular projections^{22,23}**.

We also refer the reviewer to the above comments made to Reviewers #1 and #2, who asked related questions about the parcellation. Their comments have prompted us to extend the information provided on the choice of parcellation in the Materials and Methods section (p.21).

• *L8P4: Please elaborate on the identification task, offering more details to ensure clarity and reproducibility.*

Authors' response: We thank the reviewer for this comment and have made changes to the Methods section with full specifications of the identification task (p. 18, attached to a question further down).

• *Results Section: Kindly include a reference to the complete table of results in the supplementary material for transparency and ease of access.*

Authors' response: We have now included references to the supplementary results tables in the appropriate parts of the Results section.

• *MVPA Results: Is an accuracy of 54% indicative of a strong effect? It would be helpful to compare this result with similar MVPA findings in the literature to contextualize its significance.*

Authors' response: We thank the reviewer for this comment. We have compared our results to similar chemosensory MVPA studies. While the observed effect in our stringent pre-registered leave-one-subject-out functional ROI was smaller than other studies, post-hoc analyses using anatomically defined ROIs resulted in larger and more comparable results to similar studies. We have added a sentence in the Discussion section to reflect this:

From Page 14 (Discussion):

While larger effect sizes than those in our ROI have been reported by previous chemosensory decoding studies⁹, this ROI was generated from a conservative pre-registered leave-one-subject-out protocol (see **Methods**) that prevents information leakage. An anatomically defined ROI (in **Figure 2D**) shows a stronger effect.

• *L21P8: The sentence "Conversely, a confusion matrix extracted from a spherical ROI around the IOFC peak shows that stimuli of the same flavour quality were equally likely to be confused for each other." requires clarification. Do you mean that both odor and taste stimuli associated with savory or sweet flavors were predicted by the same voxel pattern? The term confused refers to the confusion matrix but does not imply that the modality identities (odor vs. taste) were confused by the consumer—at least not solely due to activation in the OFC.*

Authors' response: We thank the reviewer for this comment. When referring to the confusion matrices, our use of the term 'confusion' or 'confused' have strictly been limited to the confusion of the decoder. We also realise that it may not be clear that the term 'flavour quality' is used to refer to the association with savouriness or sweetness across both sensory modalities, and not as a synonym for sensory modality. We have now made both clearer in the text:

From Page 11 (Results):

Conversely, a confusion matrix extracted from a **10mm** spherical ROI around the IOFC peak shows that stimuli of the same flavour quality (*i.e. the odour and the taste associated with sweetness or savouriness, respectively*) were equally likely to be confused by the decoder for each other (**Figure 4E**).

• *L30P11: Please revise the following sentence for clarity: "This phenomenon has not previously been observed in other species, as previous experiments investigating the role of the insula in processing olfactory information have found no response a univariate activation or no crossmodal encoding between familiar tastes and odours." What do you mean by in other species? If you are referring to mammals other than humans, please ensure consistency with the cited references, as the current phrasing is ambiguous.*

Authors' response: Thank you for this suggestion. We have modified the phrasing for clarity:

From Page 14 (Discussion):

This phenomenon has not previously been observed in **humans or other mammals**, as previous experiments investigating the role of the insula in processing olfactory information have found no response^{8,35}, a univariate activation^{20,21,36} or no crossmodal encoding between familiar tastes and odours^{14,32}.

• *L34P12: In the sentence "we propose that the insular patterns observed encoded identity, whereas the OFC patterns encoded value," please clarify: Do you mean that insular patterns encode flavor identity, while the OFC encodes pleasantness? Explicitly defining value in this context would improve clarity.*

Authors' response: We thank the reviewer for this comment and agree that the sentence would benefit from further clarification. While the terms 'Pleasantness' and 'Liking' are sometimes used interchangeably to describe consummatory subjective value, we prefer to use the latter term as it reduces ambiguity and ties in better with the literature on prefrontal processing of reward. We have now clarified the sentence to state:

From Page 15 (Discussion):

Based on these results, we propose that the insular patterns observed encoded identity, whereas the OFC patterns encoded **consummatory subjective** value.

• *L47P12: The term functional could be more precise; consider using electrophysiological to enhance clarity.*

Authors' response: We thank the reviewer for this comment and have revised the line accordingly.

From Page 15 (Discussion):

Anatomically, the ventral anterior insula has reciprocal projections from olfactory areas such as the piriform cortex and orbitofrontal cortex^{22,23,43}, and **electrophysiological** perturbation of these regions induces neuronal firing in the others^{44,45}

• *L47P14: Please specify the exact question posed for each task (identity, pleasantness, and intensity) to ensure methodological transparency.*

Authors' response: We thank the reviewer for this comment and have added more details on the tasks:

From Page 18 (Methods):

In each trial of the rating task, the participant was presented 1 ml of a stimulus and was asked to choose which visual cue it corresponded to, such that participants would learn the cue associated with each flavour. **During this task, the word 'Identify' was presented at the top of the screen, and participants could choose between the two visual cues.** This cue association allowed us to examine if participants were able to distinguish the flavours without

being primed by words such as 'sweet' or 'savory'. They subsequently rated it for pleasantness and intensity, during which one of the words 'Pleasantness' or 'Intensity' was presented at the top of the screen. Participants responded by moving a cursor on a visual analogue scale. For the pleasantness task, the scale was anchored at 'Extremely disgusting' at the bottom, 'Neither' in the middle, and 'Extremely pleasant' at the top. For the intensity task, the scale was anchored at 'Not at all intense' at the bottom and 'Extremely intense' at the top.

• L17-18P15: *Could you confirm whether the accuracy score was calculated based on successful associations between flavors and their corresponding visual cues? Clarifying this point would aid interpretation.*

Authors' response: Yes, this is correct; the mentioned required accuracy score of 85% refers to successful classification of the presented stimulus as being associated with the intended letter from the Phoenician alphabet. We have now added this information to the sentence in question to clarify this for the reader.

From Page 18 (Materials and Methods):

Participants completed up to 60 trials, although from trial 40 onwards the experiment ended once their cumulative accuracy rate for correctly identifying a given flavour stimulus with the correct visual stimulus was above 85%, thereby ensuring that the participants could sufficiently distinguish between the two flavour stimuli.

• L33P16: *Welcome appears to contain a typographical error (Wel(l)come). Please correct as needed.*

Authors' response: Thank you for this comment. We have double checked the spelling against the official SPM page hosted on the Wellcome Centre for Human Neuroimaging website (<https://www.fil.ion.ucl.ac.uk/spm/>) and found it to be consistent.

Reviewer #3 (Remarks to the Author):

Thank you to the authors for addressing most of my previous concerns. I have a few follow-up comments to help finalize the reviewing process.

Follow-up on previous Comment 2.B:

Thank you for your explanation. The swallow breath indeed allows for a maximal number of odorants to reach the nasal cavity; however, depending on the quantity of liquid in the mouth (the less, the better), odorants can reach the olfactory epithelium even without swallowing. Furthermore, regarding Maier et al.'s studies, since odor responses precede taste responses in the olfactory cortex (OC), and taste responses precede odor responses in the gustatory cortex (GC), the authors concluded that there are converging gustatory and olfactory inputs. However, they do not claim evidence of a direct anatomical connection.

In contrast, your brain connectivity data indicate connections between the piriform cortex and the insula, in the absence of top-down influence from the orbitofrontal cortex (OFC). This constitutes a compelling argument. Nevertheless, are the connections demonstrated by your connectivity analyses direct anatomical connections, or could they reflect indirect pathways? Please clarify this point.

Authors' response: We thank the reviewer for this comment. We agree that previous research investigating odour-induced responses in the insula does not claim a direct anatomical connection. Our results, however, indicate that insular representations of retronasal odour do not depend on top-down modulation from the OFC. Although our supplementary DCM analysis explicitly models neuronal activity (based on the validated haemodynamic response function) and effective connections therein, it is agnostic as to whether the effective connectivity observed is driven by direct or indirect axonal/synaptic connections¹. While our methodology does not allow us to readily disentangle if these results are driven by direct connections or indirect pathways, we point to established primate tractography highlighting bidirectional monosynaptic projections between the piriform cortex and the dysgranular and agranular insula^{2,3}. We have now made the limitation of our methodology clearer in our Discussion section:

From Page 16 (Discussion):

However, our main findings, supplemented by DCM analyses, strongly suggest that retronasal odour representations occur in the insula, combining with functional experiments showing early signalling of retronasal odour in the rodent gustatory insular cortex^{12-14,32}. While primate tractography indicates that our results were driven by monosynaptic projections between the piriform cortex and the dysgranular and agranular insula^{23,27,44}, whether these direct projections or more indirect pathways drove our findings cannot be disentangled using DCM of fMRI signals. Future functional studies in humans should consider methods with higher temporal resolution such as magneto/electroencephalography (M/EEG) to answer this question.

Page 14, Discussion:

Concerning the following sentence:

"This phenomenon has not previously been observed in humans or other mammals, as previous experiments investigating the role of the insula in processing olfactory information

have found no response^{8,35}, a univariate activation^{20,21,36} or no crossmodal encoding between familiar tastes and odours^{14,32}.”

This statement is accurate regarding fMRI studies; however, Maier et al.’s work in rodents did demonstrate involvement of the gustatory cortex in odor processing. Please consider clarifying this distinction in the text to avoid any potential confusion.

Authors’ response: We agree that there is abundant research in the Maier lab and others demonstrating GC involvement in odour processing. We thank the reviewer for pointing out the potential misinterpretation that we claimed to be the first to discover this. However, to our knowledge, our results are the first to show that ensemble encoding of retronasal odours in the GC overlaps with the specific tastes they are associated with. We have now revised the Discussion section to clarify this distinction:

From Page 14 (Discussion):

This phenomenon has not previously been observed in humans or other mammals, as previous experiments investigating the role of the insula in processing olfactory information have found no response in non-human primates^{8,35}; although involvement of the gustatory cortex in odour processing has been shown using human fMRI^{20,21,36} as well as rodent electrophysiology^{12–14,32,37}, our study was the first to show that retronasal odours share flavour-specific encoding with their associated tastes.

Terminology clarification:

The expression “consummatory subjective value” remains ambiguous. It is unclear whether it refers specifically to liking, wanting, palatability, or a combination of these dimensions. Please provide a clearer definition or justification of the term’s intended meaning within the context of your study.

Authors’ response: We thank the reviewer for pointing this out. We have now clarified that the OFC representations likely pertain to liking as defined by Berridge and Kringelbach⁴:

From Page 14 (Discussion):

Based on these results, we propose that the insular patterns observed encoded identity, whereas the OFC patterns encoded consummatory subjective value (liking⁵⁶).

Minor comments:

• *Supplementary Figure S7:*

o *Please add a brief explanation of the terms “commonalities” and “crossmodal” in the figure legend for clarity.*

o *Please also specify in the legend that “PEB” refers to parametric empirical Bayes.*

Authors’ response: We thank the reviewer for this comment. We have now added explanations for the commonalities and crossmodal halves of the figure, in addition to using the full term ‘Parametric Empirical Bayes’ in the first instance of the use of the abbreviation ‘PEB’.

References

1. Grefkes, C., Wang, L. E., Eickhoff, S. B. & Fink, G. R. Noradrenergic Modulation of Cortical Networks Engaged in Visuomotor Processing. *Cereb. Cortex* **20**, 783–797 (2010).
2. Mufson, E. J. & Mesulam, M.-M. -Marsel. Insula of the old world monkey. II: Afferent cortical input and comments on the claustrum. *J. Comp. Neurol.* **212**, 23–37 (1982).
3. Mesulam, M.-M. & Mufson, E. J. Insula of the old world monkey. III: Efferent cortical output and comments on function. *J. Comp. Neurol.* **212**, 38–52 (1982).
4. Berridge, K. C. & Kringelbach, M. L. Pleasure systems in the brain. *Neuron* **86**, 646–64 (2015).